# An Optimisation Approach for Long-Term Industrial Investment Planning

**Hür Bütün \*, Ivan Kantor and François Maréchal**

Industrial Process and Energy Systems Engineering (IPESE), École Polytechnique Fédérale de Lausanne, 1951 Sion, Switzerland; ivan.kantor@epfl.ch (I.K.); francois.marechal@epfl.ch (F.M.)

**\*** Correspondence: hur.butun@epfl.ch

**Abstract:** The industrial sector has a large presence in world energy consumption and $CO_2$ emissions, which has made it one of the focal points for energy and resource efficiency studies. However, large investments are required to retrofit existing industrial plants, which remains the largest barrier to implementing energy saving solutions. Process integration methods can be used to identify the best investments to improve the efficiency of plants, yet their timing remains to be answered using an optimisation approach. Even more critically, such decisions must also account for future investments to avoid stranded or regretted investments. This paper presents a method incorporating investment planning over long time horizons in the framework of process integration. The time horizon is included by formulating the problem using multiple investment periods. Investment planning is conducted using a superstructure approach, which permits both commissioning and decommissioning of units in the beginning of each period. The method is applied to a large case study, with an industrial cluster neighbouring an urban centre to also explore options of heat integration between industries and cities. Compared to the business-as-usual operation, optimal investment planning improves the operating cost of the system by 27% without budget constraints and 16–26% with constraints on budget and investment periods, which is reflected as an increase in net present value and a decrease in $CO_2$ emissions. In all cases, the operating cost benefits pay off the investment in less than two years. The present work is efficient in finding energy saving solutions based on the interest of industries. This method adds additional perspectives in the decision-making process and is adaptable to various time horizons, budgets and economic constraints.

**Keywords:** process integration; investment planning; long time horizons; energy efficiency; MILP

---

## 1. Introduction

Final energy use and direct $CO_2$ emissions in industry accounted for ~150 EJ and ~12 Gt in 2017, respectively, corresponding to 37% of global final energy consumption [1] and 34% of global $CO_2$ emissions [2]. Industry uses large quantities of coal and oil, consuming 60% of global coal and 28% of global oil production. Industrial final energy use has grown by 65% since 1971; consequently, $CO_2$ emissions are expected to increase 1.7 times by 2030 [3]. In the long term, this will result in a 2.7 °C increase in the global average temperature by 2100, which could be seen as an improvement compared to the business as usual scenario, prior to the Paris agreement, but is still not enough to prevent the possibility of dangerous changes in climate [4]. Thus, to confine the rise in the temperature below 2 °C, as targeted within the Paris agreement, more aggressive energy efficiency improvement strategies are needed. As one of the biggest energy consuming and $CO_2$ emitting actors, industry is in the spotlight of such strategies.

According to the IEA's efficient world scenario, industry has a potential to produce nearly twice as much value per unit of energy used compared to the current state [5]. Reaching this potential

depends on the deployment of best available techniques and energy efficiency measures on industrial sites. Energy efficiency and penetration of renewable energy technologies are two key elements toward reaching the environmental targets. Energy intensive industrial sectors such as petrochemicals, cement and steel have been subject to regulations in the past decade. Nevertheless, contributions from all industrial producers are required to achieve long-term targets, because most "easy" solutions have already been implemented and low-energy industries, such as food and textile manufacturing, represent 70% of the energy saving potentials in industry [5].

Developing countries account for 49% of the final energy used in industry, followed by developed countries with 40% [6]. This shows that improvements in the industrialised countries are important, as they are large contributors to the overall consumption and can change the state of the art for the developing countries. In general terms, the energy efficiency of an existing industrial plant or cluster can be improved following a wide variety of technical actions, including

- maintaining and/or refurbishing existing equipment to restore their efficiency;
- replacing and retiring obsolete equipment and production processes with the best available techniques; and
- using waste management measures such as insulation and sharing excess heat and material from one process to another.

These retrofitting actions come with investment, the biggest barrier to improving energy efficiency [7]. Energy efficiency investments are subject to rigorous criteria such as payback time lower than 12 months, thus they have to compete for capital and short-termism [8]. Conversely, it is often overlooked that current equipment on plants have a limited lifetime and investment would eventually be required, regardless of resistance to capital expenditures. Therefore, considering long time horizons provides investments in energy efficiency improvements better ground for competition over just replacing the equipment which reach their end of lifetime (EoL). Nevertheless, this adds another layer of complexity, as not only the question of "what to invest in", but also when to make the investment must be answered.

To answer these questions, this paper presents a novel method for simultaneous optimisation of investment planning and process integration. Section 2 covers the investment planning methods available in the literature, Section 3 illustrates the formulation and its detailed explanation, Section 4 presents the case study, Section 5 discusses the results and Section 6 draws the conclusions of this work.

## 2. State-Of-The-Art

Process integration (PI) is a domain in chemical engineering, which emerged due to the energy crisis in the 1970s and has been developed since, with the motivation of addressing environmental concerns, regulations and agreements. PI is based on mass and energy balances and aims to improve existing processes, decrease material and energy losses and reduce operating and investment costs, as well as environmental impact. The methods developed in the domain of PI can be considered in two main groups: graphical methods based on pinch analysis (PA) [9] and mathematical programming (MP) methods [10]. MP methods formulate PI in the form of mixed integer linear programming (MILP) [11] or mixed integer nonlinear programming (MINLP) [12] problems. In addition, the optimisation of several plants and industrial clusters [13] as well as single plants [14] has been addressed.

The applications of PI cover a wide range of industrial processes. Porzio et al. developed a PI method based on evolutionary algorithm for better integration of steel plants, focusing on process gases and their recovery [15]. Hansen et al. used PI to reduce the fresh water consumption of a petrochemical plant employing mathematical programming and following a set of heuristics [16]. Tilak and El-Halwagi studied the optimal integration of calcium looping in cement production, while considering potential symbiosis options with chemical plants [17]. However most PI methods, instead of addressing a specific industrial sector, are generic and focus on the configuration of the utility systems. Abikoye et al. proposed a flowsheeting superstructure to optimise the share of solar thermal

energy systems coupled with heat storage in low temperature industrial processes [18]. Elsido et al. developed a method for simultaneous optimisation of heat exchanger networks and utility system integration, such as organic Rankine cycles [19]. The problem was solved using a novel decomposition algorithm with integer cuts. Similarly, Kermani et al. developed a holistic and generic method, which solves heat and mass integration, heat exchanger network design and improves the industrial processes by organic Rankine cycle integration [20]. The method proposed in this work is based on a PI method [21], which introduces location aspects, such as heat losses and piping costs between plants. A comprehensive summary of the PI methods available in the literature can be found in [21]. The literature review in this work, therefore, focuses on investment planning approaches.

Investment planning has been applied in different fields, such as energy planning, carbon capture, urban systems planning and production of chemicals and pharmaceuticals. One of the branches of energy planning that has been extensively studied is generation expansion planning (GEP), which determines the type, siting, sizing and timing of new plant additions. Grigorios et al. developed an MILP GEP model using small periods (i.e., months), which results in better scheduling, as mid-term decisions are permitted [22]. They also included the cost of refurbishment of the existing units which helped with the problem convergence. Pereira et al. incorporated long and short time horizons in GEP [23]. Although the investment planning of renewable energy system penetration in electricity generation was carried out for a time horizon of 10 years, every year was evaluated in hourly time steps to investigate the short term impact of the investment decisions. It was concluded that high dependence on renewables increases the system's sensitivity to the seasonality of resources, which is often neglected in methods working only with yearly averages. The main gaps in GEP have been highlighted as not including the transmission system in the analysis and considering only centralised systems [22]. A long-term expansion planning method was developed by Zhang et al. [24] to optimise an energy hub, taking into account the transmission system. The objective was to find the system with the lowest cost of satisfying the hub requirements. The units considered for investment included generating units, transmission lines, natural gas furnaces and combined heat and power units. Botterud et al. proposed a stochastic dynamic optimisation model for investments in power generation embodying both centralised and decentralised decision-making [25]. Instead of minimising the total cost as most methods in literature, they maximised either investor profits or social welfare in the system. Energy planning models can be computationally expensive, especially when detailed time resolution is considered. Bakken et al. treated model complexity by dividing it into operational and investment sub-problems [26]. The operational planning model included alternative supply structures for multiple energy carriers such as electricity, natural gas, liquid natural gas, oil, biomass and district heating and their scheduling using hourly time steps. Afterwards, the planning of investment was carried out for a long time horizon using an investment model, in the form of dynamic programming.

Most of the methods present in literature use an economic objective, as the main focus is the investment. Although decreasing the cost indirectly helps reducing $CO_2$ emissions, there are a few methods explicitly targeting improvements in environmental impact. Mirzaesmaeeli et al. proposed a method to select the optimal mix of energy supply sources to meet the current and future electricity demand in Ontario, while minimising the cost of electricity [27]. The model also included constraints on $CO_2$ emissions, so that the selected power generation systems do not violate the regulations on emissions that are in place. Fripp created a multi-period stochastic linear programming model called Switch to reduce the environmental impact of power generation by choosing optimal portfolios for renewable energy deployment [28]. The model was able to decide how much capacity to build in different load zones, as well as how much power transfer capacity to install between them. Another novelty in the model was the flexibility of using existing systems or turning them off for a period of time, to decrease the operating and maintenance costs. Cristobal et al. studied $CO_2$ mitigation by $CO_2$ capture systems. They proposed a stochastic MILP model to retrofit a coal power plant and choose between buying $CO_2$ allowance and installing a $CO_2$ capture system, as well as to determine the

optimal time for investment [29]. Stochasticity was introduced with the variations in the future $CO_2$ allowance prices.

Investment planning in urban energy systems is generally carried out at two different scales, namely, building and district. Cano et al. developed an energy systems planning model for buildings to decide which technologies to install, as well as the time of the investments [30]. They considered ageing of technologies and its impact on system performance. A time horizon of 15 years with 12 monthly profiles and hourly time steps was considered. This way, variations in the availability of some technologies, such as PV, were taken into account. A district-level, multistage stochastic programming model was proposed by Lambert et al. [31] for optimal phasing of district heating networks. In the first step, the optimal selection of pipe diameters was conducted, minimising capital cost and heat losses. In the second step, the optimal deployment of district heating network pipes was determined, over a long time horizon.

Industrial applications of investment planning include areas of waste management, utility systems, process design and capacity expansion. Chakraborty et al. proposed a long-term operation and investment planning method for waste management [32]. Although the investment decisions were optimised for a five-year period, the optimal operation of the plant was carried out for another 20 years, to correctly asses the long-term impact of investment decisions. The method was extended, by introducing a dynamic view of designing optimal waste management strategies under uncertainty [33]. Wickart and Madlener developed a method to optimally choose between investing in an industrial boiler or a CHP unit and the appropriate investment time [34]. The effect of uncertainty was considered for fuel and electricity prices. It was concluded that if the operational risks are high, investors are likely to prefer a less capital-intensive option, i.e., investing in the steam boiler.

Sahinidis et al. studied a capacity expansion problem consisting of a network of existing and new processes with forecasts for prices and demands within a long range horizon [35]. They formulated the problem as a MILP model to optimise the net present value (NPV) and determine how much of each chemical is produced in each period, the capacity expansion and shut-down decisions. This model was extended, by including flexible processes, which could operate in both continuous and batch modes [36]. Norton and Grossmann further extended the method, by adding raw material flexibility on top of product flexibility [37]. Raw material flexibility included using different chemical feedstocks, as well as supplying them from different sources. Jain and Grossmann worked on long-term scheduling of tests in new product development in the pharmaceutical industry [38]. They proposed a method which considered the trade-off between greater product sales from a shorter-term test in parallel configuration and lower expected value of total cost from longer sequential tests. This was an extension of the work from Schmidt and Grossmann [39], considering resource limitations. Maravelias and Grossmann [40] combined the scheduling [38] and planning [37] efforts in the literature to predict which products should be tested and determine the detailed test schedules, production profiles and design decisions. The selection of the product portfolio was added as an additional decision variable and disjunctive programming was used to solve the problem.

The literature on investment planning has addressed a broad range of issues; however, the focus of research was directed mostly towards energy planning and expansion of electricity generation systems. Only a few methods in the literature propose methods for industrial problems, and even those consider processes as simple input–output models, neglecting detailed flows. PI offers an effective approach to such problems, incorporating heat cascade and mass balance constraints. A PI method targeting industrial investment over a long time horizon has not been proposed. The work presented in this paper combines the strength of investment planning and PI. This way, investments in industrial plants and clusters can be optimally planned, without compromising on the level of detail of the processes or energy conversion systems.

## 3. Method

The method proposed in this work is an MILP framework for simultaneous optimisation of process integration and long term investment planning (PIIP). Figure 1 illustrates a simple graphical overview of the method. The problem consists of multiple investment periods ($p \in \mathbf{P}$), each representing an opportunity to modify plant configuration for the next periods (e.g., one period representing one year in a time horizon of 20 years). Each period consists of single or multiple time steps ($t \in \mathbf{TT}_p$), which are used to divide their corresponding period into smaller time segments (e.g., seasons, months, days, etc.), representing different operational modes. Investment decisions are made at the beginning of each period and the system is operated within the boundaries of those decisions in the time steps of the period.

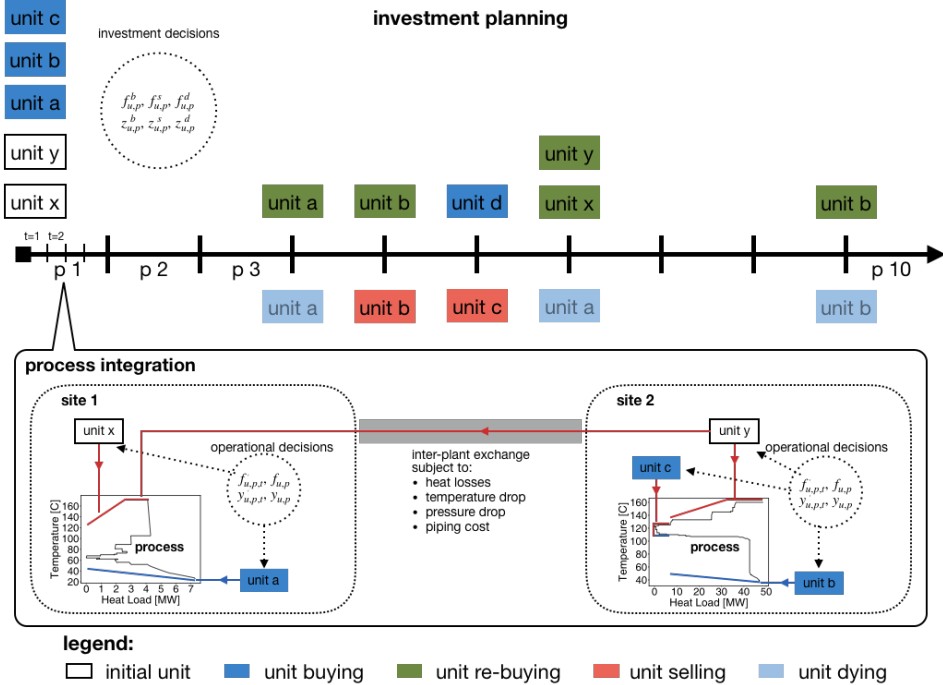

**Figure 1.** Overview of the PIIP method.

The objective function is selected as the *NPV* of the system, as given in Equation (1). *NPV* is the sum of the cash flows in the periods, discounted by the expected interest rate. Including the interest rate in the calculations makes it possible to distinguish investments in different periods.

$$\min\ NPV \tag{1}$$

$$NPV = \sum_{p \in \mathbf{P}} \left[ \frac{C_p^{cf}}{(1+\mathrm{i})^p} \right] \tag{2}$$

where $C_p^{cf}$ is the cash flow in period $p \in \mathbf{P}$ and i is the expected interest rate. Although the PI model is adapted from [21], a novel formulation for investment planning and economic analysis is proposed and integrated to PI. Thus, the main focus in this section is describing the equations governing investment planning and economic analysis. For a clear representation of the method, the PI model is discussed briefly, followed by a detailed description of the investment planning formulation and economic model.

### 3.1. Process Integration

The PI model is based on energy and resource balances. Demand and supply of energy and resources are modelled using units. The system includes two types of units in terms of their operation,

namely process units ($pu \in \mathbf{PU}$) and utility units ($uu \in \mathbf{UU}$). Process units represent manufacturing of products, and therefore have fixed size and operation, whereas utility units satisfy demands from process units and have flexible size and operation. The units in the system are clustered with respect to their locations ($lc \in \mathbf{LC}$). The heat balance is closed within each location with hot ($h \in \mathbf{HS}_{lc}$) and cold ($c \in \mathbf{CS}_{lc}$) streams from the units. Heat cascade constraints are added to ensure that heat flows from hot streams to cold streams in each temperature interval ($k \in \mathbf{K}_{lc}$), and from higher to lower temperature. Resource balances are closed within each location and for each layer ($ly \in \mathbf{L}$) representing the resource type. The electricity balance, in contrast, is closed for the overall system—simulating that all units are connected to each other through the electrical grid. Heat and resource exchanges between locations are possible, but subject to heat losses, temperature and pressure drop, and requiring the associated infrastructure. Heat sharing from a location ($lc \in \mathbf{LC}$) to another location ($ol \in \mathbf{OL}_{lc}$) can be via two different transfer types ($tr \in \mathbf{TR}$) (underground or above-ground), whereas resource sharing is assumed to take place only through underground pipes. Heat and resource stream splitting constraints ensure that heat and resource balances are not violated for inter-location exchanges. Figure 2 illustrates the main equations of the PI model. Further details on it can be found in [21].

**Sizing and scheduling: process units**

$$f'_{u,p,t} = 1 \quad \forall\, u \in \mathbf{PU}, p \in \mathbf{P}, t \in \mathbf{TT}_p$$

$$f_u = 1 \quad \forall\, u \in \mathbf{PU}, p \in \mathbf{P}$$

$$y'_{u,t} = 1 \quad \forall\, u \in \mathbf{PU}, p \in \mathbf{P}, t \in \mathbf{TT}_p$$

$$y_{u,p} = 1 \quad \forall\, u \in \mathbf{PU}, p \in \mathbf{P}$$

**Sizing and scheduling: utility units**

$$\mathrm{F}^{min}_{u,p} \cdot y_{u,p} \le f'_{u,p} \le \mathrm{F}^{max}_{u,p} \cdot y_{u,p} \quad \forall\, u \in \mathbf{UU}, p \in \mathbf{P}$$

$$f'_{u,p,t} \le f'_{u,p} \quad \forall\, u \in \mathbf{UU}, p \in \mathbf{P}, t \in \mathbf{TT}_p$$

$$\mathrm{F}^{min}_{u,p,t} \cdot y'_{u,p,t} \le f'_{u,p,t} \le \mathrm{F}^{max}_{u,p,t} \cdot y'_{u,p,t} \quad \forall\, u \in \mathbf{UU}, p \in \mathbf{P}, t \in \mathbf{TT}_p$$

$$y'_{u,p,t} \le y'_{u,p} \quad \forall\, u \in \mathbf{UU}, p \in \mathbf{P}, t \in \mathbf{TT}_p$$

**Heat cascade**

$$\left( \sum_{h \in \mathbf{HS}_{lc,k}} \dot{q}_{h,k,p,t} \cdot s_{h,p,t} \right) - \left( \sum_{c \in \mathbf{CS}_{lc,k}} \dot{q}_{c,k,p,t} \cdot s_{c,p,t} \right) - \dot{R}_{lc,k,p,t} = 0 \quad \forall\, lc \in \mathbf{LC}, k \in \mathbf{K}_{lc}, p \in \mathbf{P}, t \in \mathbf{TT}_p$$

$$\dot{R}_{lc,k,p,t} = 0 \quad \forall\, lc \in \mathbf{LC}, p \in \mathbf{P}, t \in \mathbf{TT}_p, k = first(\mathbf{K}_{lc}) \; or \; k = last(\mathbf{K}_{lc})$$

$$f'_{u,p,t} = s_{s,p,t} \quad \forall\, u \in \mathbf{U}, s \in \mathbf{H}_u, p \in \mathbf{P}, t \in \mathbf{TT}_p : s \notin \mathbf{HI}$$

**Heat stream splitting**

$$f'_{u,p,t} = \sum_{lc \in \mathbf{LC}} \sum_{tr \in \mathbf{TR}} b_{sp,lc,tr,p,t} \quad \forall\, u \in \mathbf{U}, sp \in \mathbf{SP}_u, p \in \mathbf{P}, t \in \mathbf{TT}_p$$

$$s_{s,p,t} = b_{sp,lc,tr,p,t} \quad \forall\, sp \in \mathbf{SP}, lc \in \mathbf{LC}, tr \in \mathbf{TR}, p \in \mathbf{P}, t \in \mathbf{TT}_p$$

**Resource balance**

$$\dot{m}^{in}_{ly,u,p,t} \cdot f'_{u,p,t} = \dot{M}^{in}_{u,p,t} \quad \forall\, ly \in \mathbf{L}, u \in \mathbf{U}_{ly}, p \in \mathbf{P}, t \in \mathbf{TT}_p$$

$$\dot{m}^{out}_{ly,u,p,t} \cdot f'_{u,p,t} = \dot{M}^{out}_{u,p,t} \quad \forall\, ly \in \mathbf{L}, u \in \mathbf{U}_{ly}, p \in \mathbf{P}, t \in \mathbf{TT}_p$$

$$\sum_{u \in \mathbf{U}_{ly}} \dot{M}^{in}_{u,p,t} = \sum_{u \in \mathbf{U}_{ly}} \dot{M}^{out}_{u,p,t} \quad \forall\, ly \in \mathbf{L}, p \in \mathbf{P}, t \in \mathbf{TT}_p$$

**Resource stream splitting**

$$\sum_{ol \in \mathbf{OL}_{lc}} \sum_{u \in \mathbf{U}_{ly,ol}} \dot{m}^{out}_{u,p,t} \cdot a^{out}_{ly,ol,lc,u,p,t} + \sum_{u \in \mathbf{U}_{ly,lc}} \dot{M}^{out}_{u,p,t} = \sum_{ol \in \mathbf{OL}_{lc}} \sum_{u \in \mathbf{U}_{ly,ol}} \dot{m}^{out}_{u,p,t} \cdot a^{out}_{ly,lc,ol,u,p,t} + \sum_{u \in \mathbf{U}_{ly,lc}} \dot{M}^{in}_{u,p,t} \quad \forall\, ly \in \mathbf{L}, lc \in \mathbf{LC}, p \in \mathbf{P}, t \in \mathbf{TT}_p$$

**Electricity balance**

$$\sum_{u \in \mathbf{U}} \dot{E}^{in}_{u,p,t} = \sum_{u \in \mathbf{U}} \dot{E}^{out}_{u,p,t} \quad \forall\, p \in \mathbf{P}, t \in \mathbf{TT}_p$$

**Figure 2.** Formulation of the PI problem in [21].

## 3.2. Investment Planning Model

The investment planning model consists of a set of constraints, which ensure that investment actions are logical. Such actions include commissioning and decommissioning of units as well as installation of pipes for heat and resource sharing between sites. As process units ($pu \in$ **PU**) have fixed operation, they cannot be bought or sold, which excludes them from investment analysis. The other units are classified into main groups from the investment perspective, defined as sets in the formulation:

- **BU**: The set of base case units. These units exist in the initial system in which the plants are operated business as usual. Thus, they do not need to be purchased initially.
- **NU**: The set of new units. This set consists of units that can potentially improve the efficiency of the plants (e.g., heat pumps), but currently do not exist on the sites. Therefore, they must be purchased before using them.
- **IU**: The set of investment units. This set includes base case and new units and it present to simplify the formulation, $\therefore$ **IU** $\subset$ **UU** = **BU** $\cup$ **NU**.

At the beginning of each period ($p \in$ **P**), an investment unit $u \in$ **IU** can be commissioned or decommissioned. Commissioning refers to the purchase and installation of the unit, whereas decommissioning can either reflect selling the unit or using it until the end of its lifetime. Each of these actions are modelled with binary decision variables $z^b_{u,p}$ for purchasing, $z^s_{u,p}$ for selling and $z^d_{u,p}$ for reaching EoL, respectively. For a given time horizon, these actions can happen more than once. For example, a unit can be repurchased if it is has been decommissioned at the beginning of the same period or before. It is also possible to take a commissioning and decommissioning action on the same unit in the same period. This gives flexibility to the system to repurchase units which recently reached EoL or were sold.

Investment decisions are chronological and interdependent. For instance, a new unit ($u \in$ **NU**) has to be commissioned before it is decommissioned. Another binary variable, $z^e_{u,p}$, is introduced to the problem to define units' existence and govern the relationship between the investment decisions. If a unit exists, it cannot be repurchased before decommissioning it. This also prevents progressive installation and phasing out of a unit.

A new unit ($u \in$ **NU**) exists (i.e., $z^e_{u,p} = 1$) if it has been purchased and has not yet been decommissioned. The same applies to the base case units ($u \in$ **BU**), except that they already exist in the beginning of the project. These existence constraints are imposed by Equations (3) and (4), respectively.

$$z^e_{u,p} = \sum_{pp \in \{1..p\}} (z^b_{u,pp} - z^s_{u,pp} - z^d_{u,pp}) \quad \forall\, u \in \mathbf{NU},\ p \in \mathbf{P} \tag{3}$$

$$z^e_{u,p} = 1 + \sum_{pp \in \{1..p\}} (z^b_{u,pp} - z^s_{u,pp} - z^d_{u,pp}) \quad \forall\, u \in \mathbf{BU},\ p \in \mathbf{P} \tag{4}$$

An investment unit ($u \in$ **IU**) in a period ($p \in$ **P**) can be decommissioned only if it exists in the previous period (see Equation (5)). This constraint applies to all periods except the first. In the first period, a new unit ($u \in$ **NU**) cannot be decommissioned (see Equation (6)), because it either does not exist or has just been purchased. Conversely, a base case unit can be decommissioned in the first period (see Equation (7)).

$$z^d_{u,p} + z^s_{u,p} \leq z^e_{u,p-1} \quad \forall\, u \in \mathbf{IU},\ p \in \mathbf{P} : p \neq 1 \tag{5}$$

$$z^d_{u,p} + z^s_{u,p} = 0 \quad \forall\, u \in \mathbf{NU},\ p \in \mathbf{P} : p = 1 \tag{6}$$

$$z^d_{u,p} + z^s_{u,p} \leq 1 \quad \forall\, u \in \mathbf{BU},\ p \in \mathbf{P} : p = 1 \tag{7}$$

In PI, utility units are sized according to the requirements of process units. When a utility unit is defined, it has a reference size (e.g., 100 kW boiler), which is scaled with respect to the demand, using a continuous variable, $f_{u,p}$ [21]. The same method is used to determine the real size of the investment

units; they are defined with reference sizes and scaled with continuous variables ($f$) to determine the size of the equipment that is commissioned or decommissioned. Although $f$ is literally a scaling factor, it is referred to as size in this formulation, for simplicity.

The purchase size of a unit, $f_{u,p}^b$, must be within a logical range, which reflects the minimum and maximum sizes of the technology available in the market. This is enforced by Equation (8), which also links the binary and continuous variables unit procurement.

$$z_{u,p}^b \cdot F_u^{\min} \leq f_{u,p}^b \leq z_{u,p}^b \cdot F_u^{\max} \quad \forall\, u \in \mathbf{IU},\ p \in \mathbf{P} \tag{8}$$

The available size of an investment unit ($u \in \mathbf{IU}$) changes throughout periods because of investment decisions. For example, a unit available with a certain size might be sold in a period and purchased again with a larger size in a subsequent period. A continuous variable, $f_{u,p}^e$, is introduced in the formulation to obtain the existing size of a unit in a given period. The base case units ($u \in \mathbf{BU}$) are defined with an initial size ($F_u^{\text{init}}$) according to the actual capacity of the equipment on the site, as they exist in the beginning, whereas the initial size of new units is zero (i.e., $F_u^{\text{init}} = 0 \quad \forall\, u \in \mathbf{NU}$). In the first period, the existing size is equal to the sum of the initial size and the difference between the commissioned and decommissioned sizes.

$$f_{u,p}^e = F_u^{\text{init}} + f_{u,p}^b - (f_{u,p}^s + f_{u,p}^d) \quad \forall\, u \in \mathbf{IU},\ p \in \mathbf{P} : p = 1 \tag{9}$$

where $f_{u,p}^s$ and $f_{u,p}^d$ are decommissioned sizes for selling and dying, respectively. In the other periods, the existing size is equal to the sum of what remained from the previous period and the difference between the commissioned and decommissioned sizes (Equation (10)).

$$f_{u,p}^e = f_{u,p-1}^e + f_{u,p}^b - (f_{u,p}^s + f_{u,p}^d) \quad \forall\, u \in \mathbf{IU},\ p \in \mathbf{P} : p \neq 1 \tag{10}$$

As progressive decommissioning is not allowed, the size that is phased out by decommissioning ($f_{u,p}^s$ or $f_{u,p}^d$) is equal to the size that existed before. In the first period, only the base case units ($u \in \mathbf{BU}$) can be decommissioned. Equations (11) and (12) ensure that the decommissioned size takes the value of the initial size if one of the decommissioning actions is taken.

$$f_{u,p}^s = F_u^{\text{init}} \cdot z_{u,p}^s \quad \forall\, u \in \mathbf{BU},\ p \in \mathbf{P} : p = 1 \tag{11}$$

$$f_{u,p}^d = F_u^{\text{init}} \cdot z_{u,p}^d \quad \forall\, u \in \mathbf{BU},\ p \in \mathbf{P} : p = 1 \tag{12}$$

In the other periods, the decommissioned size is equal to the existing size from the previous period. This constraint is expressed in nonlinear terms in Equations (13) and (14) and linearised in Appendix A.

$$f_{u,p}^s = f_{u,p-1}^e \cdot z_{u,p}^s \quad \forall\, u \in \mathbf{IU},\ p \in \mathbf{P} : p \neq 1 \tag{13}$$

$$f_{u,p}^d = f_{u,p-1}^e \cdot z_{u,p}^d \quad \forall\, u \in \mathbf{IU},\ p \in \mathbf{P} : p \neq 1 \tag{14}$$

A unit can be used only as long as its lifetime. The remaining life ($l_{u,p}$) is defined as an integer variable which also depends on investment decisions. The constraints given in Equations (15)–(21) govern the relationship between the unit life and the rest of the formulation:

- A unit can exist only if it has a remaining life (Equation (15)).
- Only the existing units have a remaining life (Equation (16)).
- In the first period, the remaining life is equal to either the life span (for new units) or the difference between the life span and the initial age (base case units) (Equation (17)).
- In the other periods, the remaining life decreases compared from the previous period by one period. In addition, buying actions increase the remaining lifetime while selling decreases it (Equation (18)).
- A unit can be purchased again only after it is decommissioned (Equation (19)).

- A unit dies only if its lifetime in the previous period is one year (Equation (20)).
- A unit can be sold only if its lifetime in the previous period is two years or more (Equation (21)).

$$z_{u,p}^e \leq l_{u,p} \quad \forall\, u \in \mathbf{IU},\ p \in \mathbf{P} \tag{15}$$

$$l_{u,p} \leq z_{u,p}^e \cdot \mathrm{LI}_u^{\mathrm{lt}} \quad \forall\, u \in \mathbf{IU},\ p \in \mathbf{P} \tag{16}$$

$$l_{u,p} = \left(z_{u,p}^b \cdot \mathrm{LI}_u^{\mathrm{lt}}\right) + \left(\mathrm{LI}_u^{\mathrm{lt}} - \mathrm{LI}_u^{\mathrm{init}}\right) \cdot \left(1 - z_{u,p}^s - z_{u,p}^d\right) \quad \forall\, u \in \mathbf{IU},\ p \in \mathbf{P} : p = 1 \tag{17}$$

$$l_{u,p} = l_{u,p-1} - z_{u,p-1}^e + \left(\mathrm{LI}_u^{\mathrm{lt}} \cdot z_{u,p}^b\right) - l_{u,p}^s \quad \forall\, u \in \mathbf{IU},\ p \in \mathbf{P} : p \neq 1 \tag{18}$$

$$l_{u,p-1} - l_{u,p}^s \leq \left(1 - z_{u,p}^b\right) \cdot \mathrm{LI}_u^{\mathrm{lt}} + 1 \quad \forall\, u \in \mathbf{IU},\ p \in \mathbf{P} : p \neq 1 \tag{19}$$

$$l_{u,p-1} \leq \left(1 - z_{u,p}^d\right) \cdot \mathrm{LI}_u^{\mathrm{lt}} + 1 \quad \forall\, u \in \mathbf{IU},\ p \in \mathbf{P} : p \neq 1 \tag{20}$$

$$\left(1 - z_{u,p-1}^e\right) \cdot \mathrm{LI}_u^{\mathrm{lt}} + \left(1 - z_{u,p-1}^s\right) \cdot \mathrm{LI}_u^{\mathrm{lt}} + l_{u,p-1} \leq 2 \quad \forall\, u \in \mathbf{IU},\ p \in \mathbf{P} : p \neq 1 \tag{21}$$

where $\mathrm{LI}_u^{\mathrm{lt}}$ is the unit life span, $\mathrm{LI}_u^{\mathrm{init}}$ is the initial age and $l_{u,p}^s$ is the life of the unit at the period it is sold. $l_{u,p}^s$ is equal to the remaining life of the unit if it is sold, and zero otherwise. This is ensured by Equations (22)–(24):

$$l_{u,p}^s \geq l_{u,p-1} - z_{u,p-1}^e - \left(1 - z_{u,p-1}^s\right) \cdot \mathrm{LI}_u^{\mathrm{lt}} \quad \forall\, u \in \mathbf{NU},\ p \in \mathbf{P} : p \neq 1 \tag{22}$$

$$l_{u,p}^s \leq l_{u,p-1} - z_{u,p-1}^e \quad \forall\, u \in \mathbf{NU},\ p \in \mathbf{P} : p \neq 1 \tag{23}$$

$$l_{u,p}^s \leq z_{u,p-1}^s \cdot \mathrm{LI}_u^{\mathrm{lt}} \quad \forall\, u \in \mathbf{NU},\ p \in \mathbf{P} : p \neq 1 \tag{24}$$

A unit can be used only if it exists and as much as its existing size. Equations (25) and (26) impose such existence constraints and connect the investment planning model with PI.

$$y_{u,p} \leq z_{u,p}^e \quad \forall\, u \in \mathbf{IU},\ p \in \mathbf{P} \tag{25}$$

$$f_{u,p} \leq f_{u,p}^e \quad \forall\, u \in \mathbf{IU},\ p \in \mathbf{P} \tag{26}$$

where $y_{u,p}$ is a binary decision variable for whether a unit is used or not and $f_{u,p}$ is a continuous decision variable reflecting the used capacity. Investment planning constraints for heat and resource sharing pipes are similar to those for units, though with a few added constraints to reflect industrial reality. Pipelines are long-lasting and, once installed, are used until the end of their useful service. The formulation for pipelines therefore eliminates the possibility of decommissioning and the lifetime is considered to extend beyond the planning horizon. Thus, investment decisions on pipes can be reduced to a decision on procurement alone. Detailed equations governing the investment planning for pipes are given in Appendix A.

### 3.3. Economic Model

The economic model comprises constraints to calculate cash flows and thus serves as a link between the investment planning model and the objective function. At the beginning of each period, investment actions are taken to either commission or decommission units and purchase pipes for heat and resource sharing between sites. Investment in units and pipes is considered as negative cash flow, whereas decommissioning actions are reflected as positive cash flow, as even at EoL, units retain some monetary value (i.e., scrap value). In addition, units are operated during each period, consuming resources, such as natural gas and electricity, which are reflected as negative cash flow. With retrofit investments within and between sites, the current operating bill is reduced which is considered as a positive cash flow. The net cash flow in a given period is calculated by summing the positive and negative flows as in Equation (27).

$$C_p^{cf} = \left( C_p^s + C_p^{sc} - C_p^{inv} \right) + \left( c^{\text{op}_{\text{cur}}} - C_p^{op} \right) \quad \forall\, p \in \mathbf{P} \tag{27}$$

where $C_p^s$ and $C_p^{sc}$ represent income from selling units and scrap, $C_p^{op}$ and $C_p^{inv}$ are the investment and operating costs and $c^{\text{op}_{\text{cur}}}$ is the current operating bill without any energy efficiency improvement. The operating cost is calculated using Equation (28) accounting for fixed cost (e.g., maintenance) associated with the activation of the units and variable cost associated with the unit sizes.

$$C_p^{op} = \sum_{u \in \mathbf{U}} \left[ \sum_{t \in \mathbf{TT}_p} \left( c_u^{\text{op1}} \cdot y_{u,p} + c_u^{\text{op2}} \cdot f_{u,p} \right) \cdot \Delta t_t^{\text{op}} \right] \quad \forall\, u \in \mathbf{U},\ p \in \mathbf{P} \tag{28}$$

where $c_u^{\text{op1}}$ and $c_u^{\text{op2}}$ are fixed and variable operating costs and $\Delta t_t^{\text{op}}$ is the operating time. According to the guidelines suggested in [41], the investment cost of a unit corresponds to the bare module cost, which comprises the purchase cost of the equipment, materials (e.g., fittings), labour, freight, overhead and engineering costs. For piping cost, a function including trenching is used [21]. Details of the piping economic calculations are given in Appendix A. Equation (29) is used to calculate the total investment cost in a given period. The investments are, when applicable, constrained with overall and annual budget limits which is explained in detail in Appendix A as well.

$$C_p^{inv} = \sum_{u \in \mathbf{IU}} \left( C_{u,p}^b + C_{u,p}^{mt} + C_{u,p}^{lr} C_{u,p}^{fr} + C_{u,p}^{oh} + C_{u,p}^{en} \right) + C_p^{ph} + C_p^{pr} \quad \forall\, p \in \mathbf{P} \tag{29}$$

where $C_{u,p}^b$, $C_{u,p}^{mt}$, $C_{u,p}^{lr}$, $C_{u,p}^{fr}$, $C_{u,p}^{oh}$ and $C_{u,p}^{en}$ are the purchasing, materials, labor, freight, overhead and engineering costs of the units, respectively, and $C_p^{ph}$ and $C_p^{pr}$ are heat and resource piping costs, respectively. Purchase cost is calculated based on the investment decisions $z_{u,p}^b$ and $f_{u,p}^b$ in Equation (30). All the other components of the bare module cost are calculated as a fraction of the purchase cost in Equations (31)–(35). In the case of re-buying a unit, although investment on the equipment itself, labour, freight and overhead is required again, reinvesting in materials and engineering can be avoided. Thus, materials and engineering costs apply only to new units, $u \in \mathbf{NU}$, when they are purchased for the first time.

$$C_{u,p}^b = c_u^{\text{inv1}} \cdot z_{u,p}^b + c_u^{\text{inv2}} \cdot f_{u,p}^b \quad \forall\, u \in \mathbf{IU},\ p \in \mathbf{P} \tag{30}$$

$$C_{u,p}^{mt} = C_{u,p}^b \cdot \mathrm{F}_u^{\text{mt}} \cdot \left( 1 - z_{u,p}^{bb} \right) \quad \forall\, u \in \mathbf{NU},\ p \in \mathbf{P} \tag{31}$$

$$C_{u,p}^{lr} = C_{u,p}^b \cdot \mathrm{F}_u^{\text{lr}} \quad \forall\, u \in \mathbf{IU},\ p \in \mathbf{P} \tag{32}$$

$$C_{u,p}^{fr} = C_{u,p}^b \cdot \mathrm{F}_u^{\text{fr}} \quad \forall\, u \in \mathbf{IU},\ p \in \mathbf{P} \tag{33}$$

$$C_{u,p}^{oh} = C_{u,p}^b \cdot \mathrm{F}_u^{\text{oh}} \quad \forall\, u \in \mathbf{IU},\ p \in \mathbf{P} \tag{34}$$

$$C_{u,p}^{en} = C_{u,p}^b \cdot \mathrm{F}_u^{\text{en}} \cdot \left( 1 - z_{u,p}^{bb} \right) \quad \forall\, u \in \mathbf{NU},\ p \in \mathbf{P} \tag{35}$$

where $\mathrm{F}_u^{\text{mt}}$, $\mathrm{F}_u^{\text{lr}}$, $\mathrm{F}_u^{\text{fr}}$, $\mathrm{F}_u^{\text{oh}}$ and $\mathrm{F}_u^{\text{en}}$ are cost factors for materials, labour, freight, overhead and engineering, respectively; $c_u^{\text{inv1}}$ and $c_u^{\text{inv2}}$ are fixed and variable investment cost parameters related to the existence and size of the units, respectively; and $z_{u,p}^{bb}$ is a binary variable which is activated if a unit has been previously purchased. Although the cost factors are adapted from [41], the investment cost parameters are derived using equipment cost functions. Equations (31) and (35) are nonlinear equations, replaced by a set of linear constraints; the details of which are given in Appendix A. The binary variable $z_{u,p}^{bb}$ takes the value 1 if its corresponding unit has been purchased in one of the previous periods and 0 otherwise. This is ensured by Equations (36)–(38).

$$z_{u,p}^{bb} = 0 \quad \forall\, u \in \mathbf{NU},\ p \in \mathbf{P} : p = 1 \tag{36}$$

$$z_{u,p}^{bb} \leq \sum_{pp \in \{1..p-1\}} z_{u,pp}^{b} \quad \forall\, u \in \mathbf{NU},\ p \in \mathbf{P} : p \neq 1 \tag{37}$$

$$z_{u,p}^{bb} \geq z_{u,pp}^{b} \quad \forall\, u \in \mathbf{NU},\ p \in \mathbf{P},\, pp \in \{1..p-1\} : p \neq 1 \tag{38}$$

After a unit is purchased, it starts to lose its economic value. A double declining depreciation method is used in this work, as it is more realistic compared to straight line depreciation [41]. In the first period, only the base case units have remaining value, whereas this value is zero for new units (Equations (39) and (40)). In the other periods, remaining value and depreciation are calculated with respect to each other and the investment decisions (see Equations (41) and (42)).

$$C_{u,p}^{rv} = c_u^{\mathrm{b}} \cdot \left(1 - 2 \cdot r_u^{\mathrm{dep}}\right)^{\mathrm{LI}_u^{\mathrm{init}}} \quad \forall\, u \in \mathbf{BU},\ p \in \mathbf{P} : p = 1 \tag{39}$$

$$C_{u,p}^{rv} = 0 \quad \forall\, u \in \mathbf{NU},\ p \in \mathbf{P} : p = 1 \tag{40}$$

$$C_{u,p}^{rv} = \left(C_{u,p-1}^{b} - C_{u,p-1}^{sv} - C_{u,p-1}^{dv}\right) + C_{u,p-1}^{rv} - C_{u,p-1}^{dep} \quad \forall\, u \in \mathbf{IU},\ p \in \mathbf{P} : p \neq 1 \tag{41}$$

$$C_{u,p}^{dep} = \left[\left(C_{u,p}^{b} - C_{u,p}^{sv} - C_{u,p}^{dv}\right) + C_{u,p}^{rv}\right] \cdot r_u^{\mathrm{dep}} \quad \forall\, u \in \mathbf{IU},\ p \in \mathbf{P} \tag{42}$$

$C_{u,p}^{rv}$, $C_{u,p}^{dep}$, $C_{u,p}^{sv}$ and $C_{u,p}^{dv}$ are the remaining, depreciated, sold and EoL values, respectively; $c_u^{\mathrm{b}}$ is the purchase cost of the base case units; and $r_u^{\mathrm{dep}}$ is the depreciation rate. $C_{u,p}^{dv}$ and $C_{u,p}^{sv}$ are continuous variables that take the remaining value of the unit if it reaches EoL or is sold, respectively. The relationship between them and the remaining value is enforced by Equations (43) and (44). The conversion of these nonlinear equations into a set of linear constraints is explained in Appendix A.

$$C_{u,p}^{sv} = C_{u,p}^{rv} \cdot z_{u,p}^{s} \quad \forall\, u \in \mathbf{IU},\ p \in \mathbf{P} \tag{43}$$

$$C_{u,p}^{dv} = C_{u,p}^{rv} \cdot z_{u,p}^{d} \quad \forall\, u \in \mathbf{IU},\ p \in \mathbf{P} \tag{44}$$

It is assumed that if a unit reaches EoL it retains its salvage value ($c_u^{\mathrm{sal}}$), which is typically a small fraction of the initial investment (see Equation (45)). Conversely, if it is sold before reaching EoL, the remaining value is the maximum of $C_{u,p}^{sv}$ and the salvage value, as given in Equation (46). The maximum function is nonlinear; however, it can be converted to a set of linear equations, as explained in Appendix A.

$$C_{u,p}^{sc} = c_u^{\mathrm{sal}} \cdot z_{u,p}^{d} \quad \forall\, u \in \mathbf{IU},\ p \in \mathbf{P} \tag{45}$$

$$C_{u,p}^{s} = \max\left(C_{u,p}^{sv}, c_u^{\mathrm{sal}}\right) \quad \forall\, u \in \mathbf{IU},\ p \in \mathbf{P} \tag{46}$$

### 3.4. Solution Strategy

The MILP model presented in this work can be solved by commercial solvers such as Gurobi [42] or Cplex [43], using a linear programming-based branch and bound algorithm. However, if a large industrial case study with several plants is considered, the problem size increases drastically. This increase is related to the large number of units and to the number of potential connections between plants for heat and resource sharing purposes. Taking into account these aspects within a multi-period formulation considering a long time horizon makes the model computationally expensive, even simply for finding an integer feasible solution. To solve large-scale problems without compromising model complexity, a solution strategy is proposed. Model testing identified piping between the plants to be the bottleneck. This can be explained due to the variety of heat and resource sharing media (e.g., steam at different pressure levels), directions (i.e., several candidates for excess heat) and modes (i.e., underground and above-ground). The suggested solution strategy solves the problem by initially neglecting plant connections, which provides a feasible integer solution. With this incumbent solution,

the larger problem becomes tractable and can be solved to optimality within a shorter time frame, as a result of reduced computational burden. Figure 3 schematically illustrates the solution strategy.

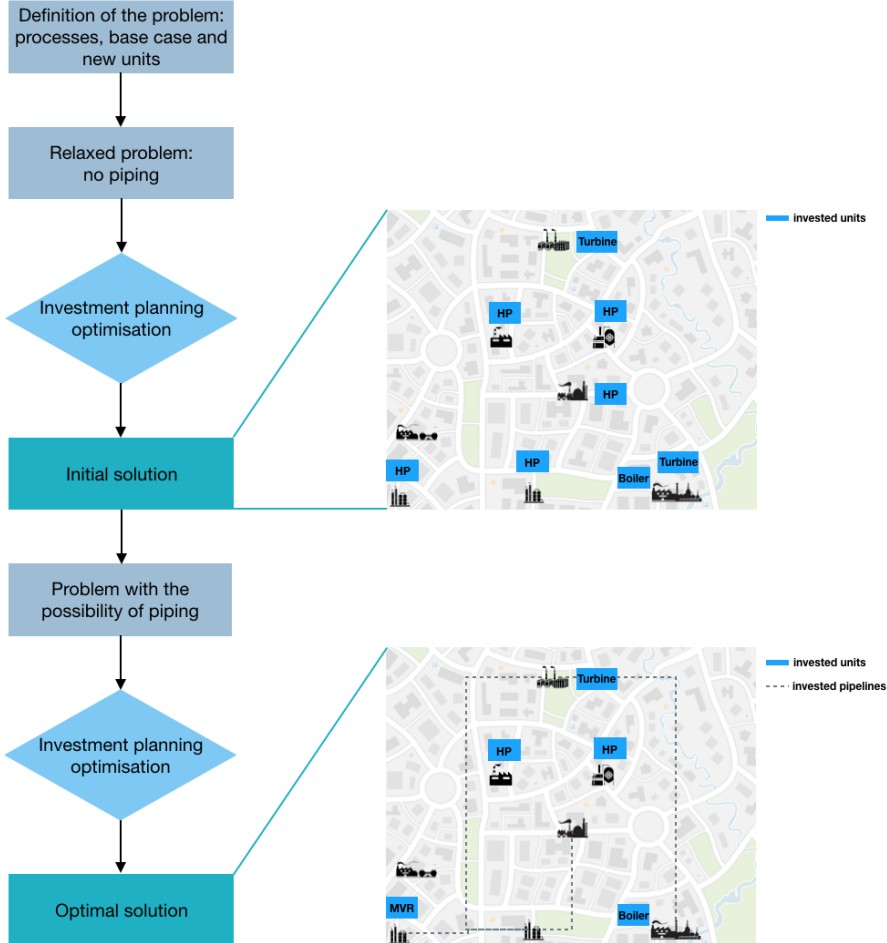

**Figure 3.** Strategy for solving the problem in two steps; initialisation and optimisation.

### 3.5. Systematic Generation of Multiple Solutions

Finding a single optimal solution in real industrial problems may be problematic, as there are often practical constraints that cannot be accounted for in the mathematical programming framework. In such cases, it is beneficial to provide multiple solutions for industries to select that which best fits their interest. Parametric optimisation is a technique used to generate multiple solutions in a systematic way aiming at optimising more than one objective function (see Equation (47)) [44].

$$\min\ f(x,y), g(x,y) \tag{47}$$

The multi-objective optimisation problem is reformulated, such that one of the objective function is optimised while the other one is constrained (see Equations (48) and (49)) above or below certain parameters (i.e., $\epsilon$), which are increased or decreased systematically, resulting in a pool of optimal solutions.

$$\min\ f(x,y) \tag{48}$$

$$g(x,y) \leq \epsilon \tag{49}$$

Although the solution strategy presented in Section 3.4 decreases the computation time, it is not sufficient for parametric optimisation in which several optimisation runs are carried out. To solve the problem effectively and generate multiple interesting solutions, a different strategy is followed.

First, the parametric optimisation problem is solved using the method in [21], without considering investment planning, setting the objective as the sum of the annual operating and annualised utility investment costs while the annualised piping cost is constrained with $\epsilon$. This results in an initial solution pool with investment targets on piping between the plants as well as energy conversion technologies. Based on those targets, the binary variables to invest in pipes and units are fixed, as well as the sizes of the pipes, and PIIP is solved for each solution in the initial pool to determine the optimal timing for investments considering yearly and overall investment budgets. Figure 4 depicts the parametric optimisation solution strategy.

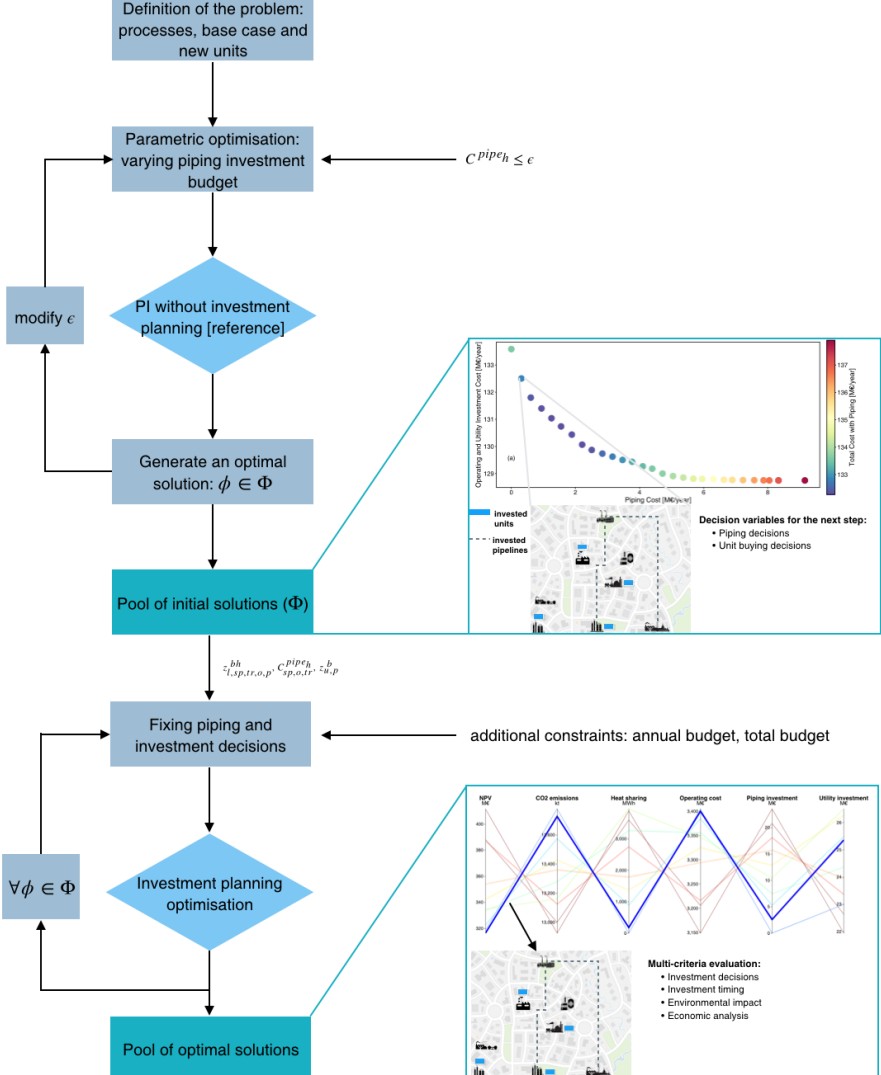

**Figure 4.** Parametric optimisation solution strategy: targeting and optimising.

## 4. Case Studies and Utility Systems

The case study is adapted from the authors of [21] and consists of nine locations. In eight of the locations there are industrial plants operating at their business as usual state, while a district is placed in one of the locations, representing part of a city close to the corresponding industrial cluster. Energy and resource balances are closed within each location at the current state. Thus, all locations have access to the resources required for their operations (e.g., natural gas and electricity) as well as energy conversion systems (e.g., boiler) to provide the required services. As the focus of this work is energy consumption, the industrial plants and the district are modelled only using their energy

flows, i.e., their electricity, and hot and cold streams. The resources considered are, therefore, linked to provision of energy services, such as natural gas, electricity and water.

The models of the industrial plants are adapted from [45,46] and scaled with the flowrate of the main product. The district model includes the demand for district services, such as space heating, domestic hot water, cooling, refrigeration and electricity, representing a potential symbiosis with the industry by heat sharing. This model is adapted from [47] and scaled with a population of 50,000 people, representing a typical medium-sized district. Figure 5 illustrates the overview of the case study with the locations and sizes of the sites.

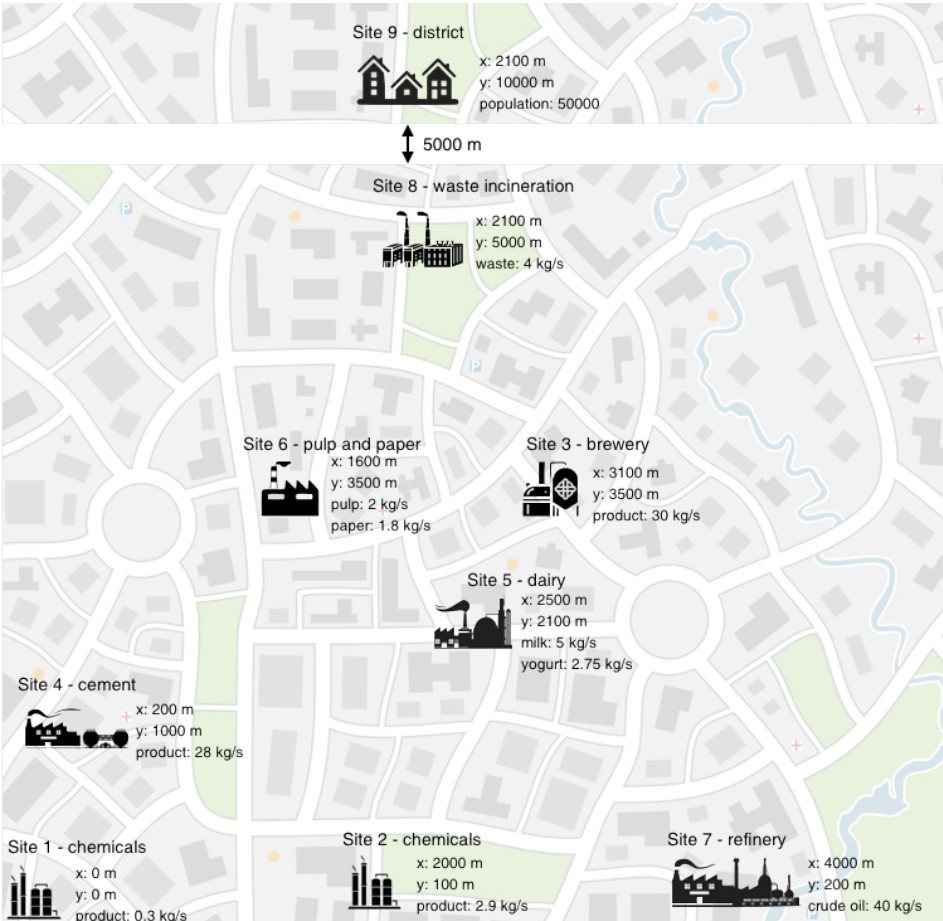

**Figure 5.** Layout of the industrial cluster neighbouring a district.

Assuming consistent industrial production, the industrial plants are modelled with fixed production rates. In addition, production capacity expansion throughout the studied time horizon is not considered. Seasonal variations are considered in the district model, as the demand for the district services change drastically throughout a year. The population of the district is assumed to remain constant during the evaluated project time. A time horizon of 20 years is evaluated with each year corresponding to a period in the mathematical formulation.

### 4.1. Utility Systems and Resources

Utility systems include energy conversion technologies that currently exist on the sites, $u \in \mathbf{BU}$, and the ones which can be integrated to improve the system, $u \in \mathbf{NU}$. The utility systems of all the sites are included in the mass and energy balance analysis, but several are excluded from investment planning:

- Site 4: The heating utility of the cement site is kiln, whereas cooling is carried out by air. Recovering the excess heat from the cement site and using it in other sites is not considered, as the technologies required are not mature enough. Thus, improvements in the utility system of the cement plant is not included in investment planning.
- Site 8: The waste incineration plant has a symbiosis potential by heat sharing with the other sites; however, improving the plant itself by integrating more efficient technologies is not studied in this work.
- Site 9: The district model is added in the case study to extend the potential of symbiosis. Except for sharing industrial excess heat with the district using a pipeline, improvements in the district utility system are not examined.

### 4.1.1. Existing Technologies on the Plants

Each plant is currently operated with conventional energy conversion technologies, such as boilers, steam networks, cooling circuits and towers. The existing technologies have the advantage that the investment has already been made, and therefore do not require an initial investment. However, they are aged equipment and are often less efficient than the competitors available in the market (alternative technologies or more efficient, modern replacements).

### Boilers

Boilers are the most common technology used in industrial plants to convert chemical energy into heat by combustion. All industrial plants in the case study, except for cement, have boilers, which currently supply their heating requirements. The boilers in this case study are modelled according to the guidelines suggested in [11]. Table 1 depicts the investment parameters ($c^{inv1}$, $c^{inv2}$) as well as the initial size ($F^{init}$) and age ($LI^{init}$) of the boilers. The fixed and variable investment costs are calculated according to [48] and the life span is considered as 20 years, according to the authors of [49].

**Table 1.** Existing boilers and their investment parameters.

| Location | $F^{init}$ (-) | $c^{inv1}$ (k€) | $c^{inv2}$ (k€) | $LI^{init}$ (years) |
|---|---|---|---|---|
| Site 1 | 7 | 388 | 13 | 16 |
| Site 2 | 62 | 388 | 13 | 12 |
| Site 3 | 30 | 388 | 13 | 11 |
| Site 5 | 19 | 388 | 13 | 9 |
| Site 6 | 11 | 388 | 13 | 6 |
| Site 7 | 190 | 388 | 13 | 8 |

### Steam Networks

Steam networks are used to distribute high temperature heat generated in boilers to the processes on site. Distributing heat using a steam network is advantageous, not only because steam is a good heat transfer fluid but also as electricity is co-generated by expanding high-pressure steam through turbines. The steam network model of each site is built as a super-structure, following the method of [50].

Steam networks consist of turbines and steam production and distribution levels, called headers, which are simply pipelines. As the pipelines are already installed on the sites and have a long lifetime, it is assumed that only the turbines are involved in the investment planning decisions. The existing turbines and their investment parameters are depicted in Table 2. The investment cost parameters are calculated according to the method in [41] and linearised to fit the MILP framework. The life span of turbines is assumed to be 20 years [49].

**Table 2.** Existing steam network turbines and their investment parameters.

| Location | Inlet (bar) | Outlet (bar) | $F^{init}$ (-) | $c^{inv1}$ (k€) | $c^{inv2}$ (k€) | $LI^{init}$ (years) |
|---|---|---|---|---|---|---|
| Site 1 | 45 | 24 | 1.3 | 64 | 22 | 16 |
| Site 1 | 45 | 8 | 1.6 | 153 | 19 | 16 |
| Site 2 | 45 | 24 | 12.2 | 64 | 22 | 12 |
| Site 2 | 45 | 8 | 12.4 | 153 | 19 | 12 |
| Site 3 | 45 | 2 | 13 | 232 | 16 | 11 |
| Site 5 | 45 | 2 | 8 | 232 | 16 | 9 |
| Site 6 | 45 | 4 | 5 | 195 | 18 | 6 |
| Site 7 | 45 | 24 | 24 | 64 | 22 | 8 |
| Site 7 | 45 | 8 | 14 | 153 | 19 | 8 |
| Site 7 | 45 | 4 | 50 | 195 | 18 | 8 |

Cooling Towers

The main cooling media in industrial plants are air and water. While heat from processes is discharged to the environment directly from aero-coolers, cooling water circuits first collect the excess heat in water and then release it to the environment via cooling towers. The cooling tower model in this work is adapted from that in [51]. Table 3 outlines the investment parameters of the cooling towers in the system. The life span of cooling towers is estimated to be 25 years [49] and the investment cost parameters are calculated according to the method in [48].

**Table 3.** Existing cooling and their investment parameters.

| Location | $F^{init}$ (-) | $c^{inv1}$ (k€) | $c^{inv2}$ (k€) | $LI^{init}$ (years) |
|---|---|---|---|---|
| Site 1 | 9 | 82 | 13 | 15 |
| Site 2 | 57 | 82 | 13 | 10 |
| Site 3 | 22 | 82 | 13 | 3 |
| Site 5 | 5 | 82 | 13 | 14 |
| Site 6 | 2 | 82 | 13 | 6 |
| Site 7 | 150 | 82 | 13 | 4 |

4.1.2. Additional Technologies

Energy conversion technologies that can potentially improve the efficiency and operating cost of the system are considered as additional technologies. Although they are more efficient than the technologies already installed on the plants, they require investment, which might pose a barrier to their purchase and installation. Appropriate additional technologies are selected based on the grand composite curves (GCCs) of the plants given in Appendix A.

Heat Pumps

Heat pumps (HPs) are used to recover low temperature excess heat and upgrade it to a higher temperature. Site 1, 2, 5 and 7 have a potential for HP integration, as they have a pinch temperature at which HPs can operate and heat recovery is possible with a small temperature lift. The investment cost of HPs is calculated according to the method in [41], considering that the main contributors are two heat exchangers (i.e., evaporator and condenser) and a compressor. The life span of the HPs is estimated as 15 years, according to the method in [49]. Table 4 summarises the investment parameters of the HPs.

**Table 4.** Potential heat pumps and their investment parameters.

| Location | $c^{inv1}$ (k€) | $c^{inv2}$ (k€) | $LI^{lt}$ (years) |
|---|---|---|---|
| Site 1 | 26 | 52 | 15 |
| Site 2 | 556 | 216 | 15 |
| Site 5 | 270 | 454 | 15 |
| Site 7 | 305 | 217 | 15 |

Mechanical Vapor Recompression

Mechanical vapour recompression (MVR) works using a similar principle to HPs, but instead of using an intermediate fluid, vapor is compressed to a higher pressure and temperature. In this case study, sites 1, 3, and 6 have a potential for MVR integration when importing 1 bar steam from the other sites and upgrading it to 2 bar steam. The investment parameters of the MVRs are calculated according to [41], and the life span is estimated as 15 years [49]. Table 5 shows the investment parameters of the MVRs.

**Table 5.** Potential mechanical vapour recompression and their investment parameters.

| Location | $c^{inv1}$ (k€) | $c^{inv2}$ (k€) | $LI^{lt}$ (years) |
|---|---|---|---|
| Site 1 | 36 | 317 | 15 |
| Site 3 | 151 | 261 | 15 |
| Site 6 | 9 | 38 | 15 |

Internal Combustion Engines

Internal combustion engines (ICEs) are alternatives to industrial boilers. They have the advantage of co-generating heat and electricity. However, because of engine cooling water, they are applicable only for processes with low pinch point. In addition, they are not used for large scale applications. Hence, they can only partially replace boilers. Based on the preliminary analysis of the GCCs, sites 1, 3, 5 and 6 have a potential for ICE integration. The investment cost parameters of the engines are adapted from those in [46] as 117 k€ and 1169 k€ for the fixed and variable cost, respectively. The life span is estimated as 20 years [49].

## 5. Results and Discussion

The method is applied to the case study following several scenarios and solution strategies. Section 5.1 determines investment planning without limitation on the budget, Section 5.2 studies the impact of seasonality in the investment decisions, Section 5.3 considers restricting the investment budget as well as the investment period and Section 5.4 considers parametric optimisation to obtain multiple investment scenarios. Section 5.5 compares the solutions in Sections 5.1 and 5.3 with the business as usual operations and investments of the industrial cluster.

### 5.1. Optimal Investment Decisions without Budget Constraints

Optimal investment planning for the system introduced in Section 4 is determined for a horizon of 20 years, without any budget constraints. The optimal NPV is obtained as 463 M€, considering operating and investment costs, resulting in 7748 kt savings on $CO_2$ emissions. The investment decisions can be grouped in two; within the plants on energy conversion systems and between the plants on piping. The investment cost in the optimal solution totals 107 M€, dominated by investments in infrastructure within the plants, which represent 79%. Figure 6 depicts the results in terms of investment cost and the year of investment; to maintain simplicity and clarity, decommissioning is not included in the figure.

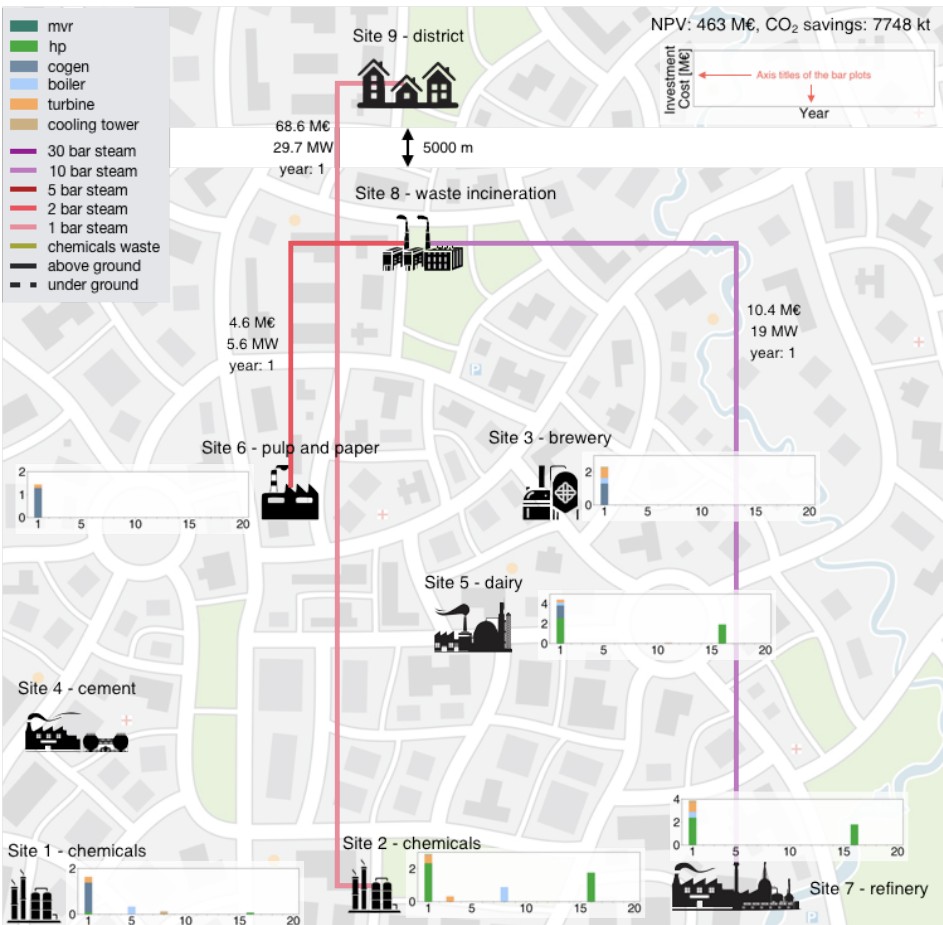

**Figure 6.** Optimal investment planning without budget constraints.

Cogeneration engines are installed in Sites 1, 3, 5 and 6 as these sites have relatively low pinch points and thus allow such integration. In addition, heat pumps are integrated in Sites 1, 2 and 7, taking advantage of transferring heat across the pinch point with a small temperature lift. As heat pumps have a life span of 15 years, they reach their EoL before the end of the evaluated project period. For this reason, recurring investment is be observed; this also implies that their payback time is less than five years.

In addition to the integration of more efficient energy conversion technologies, the system is improved by installing steam pipes between the sites. Heat is shared using high-pressure steam (e.g., 10 bar) from Site 8 to 7 and low-pressure steam (e.g., 2 bar) from Site 8 to 6 and Site 2 to 9. Although Site 9 represents a heat sharing option with a longer distance compared to the other plants, it is still selected in the optimal solution, as the energy prices are higher for the district compared to the industries. Thus, replacing a district boiler with excess heat from the industry is more profitable than replacing an industrial boiler. Site 2 is selected as the main source to provide heat to Site 9, even though the distance is greater than to Site 8, due to economies of scale (i.e., more heat is available at Site 2 compared to Site 8) and as the heat from Site 8 is at higher temperature and can be used for other plants. A Similar phenomenon is observed in the distribution of heat from Site 8 to the other industrial sites; instead of multiple neighbouring sites (e.g., Site 3 and 5), heat is shared with Site 7, as it requires a higher amount, but installing only one pipeline.

Chronology of investments show that most occur in the first period. This is logical as investments yielding economic benefits should be made as soon as possible to take full advantage over the planning horizon. The few investments made in subsequent periods are replacements for equipment reaching their EoL. Investment in boilers in Site 1 and 2 are examples of such decisions. However, the boilers in Site 3 and 7 are repurchased in the first period, which might be related to the age and size of the

equipment, i.e., as they are currently oversized, selling them before further ageing is more profitable for the system. However, as the plants would still need heating utilities after the existing boilers are sold, new ones are purchased in the first period. The piping investment decisions, similar to the equipment investment, are taken as early as possible, to benefit from corresponding operational savings.

### 5.2. Impact of Seasonality in the Investment Decisions

The impact of seasonality in investment planning is studied by considering four seasons (i.e., time steps) in a 20-year time horizon (i.e., periods), as seen in Figure 7. As stated in Section 4, only the district demand changes seasonally, which is reflected as a slight decrease in NPV. The most drastic change occurs in the piping investment decisions. As the district has higher demand in winter compared to the annual average considered in Section 5.1, the amount of steam transferred from Site 2 increases, even though the piping investment stays the same as the pipe size is large enough to handle a higher flowrate. In addition, all excess heat available on Site 8 is transferred to Site 9 and Site 2. In winter, the heat is wholly transferred to the district in the form of low pressure steam; whereas, in summer, it is shared with Site 2 as high-pressure steam, as the district heating demand is very small in summer and the chemical site has a constant demand throughout the year. In the other seasons, the excess heat from Site 8 is shared between the district and the chemical site, giving priority to the district.

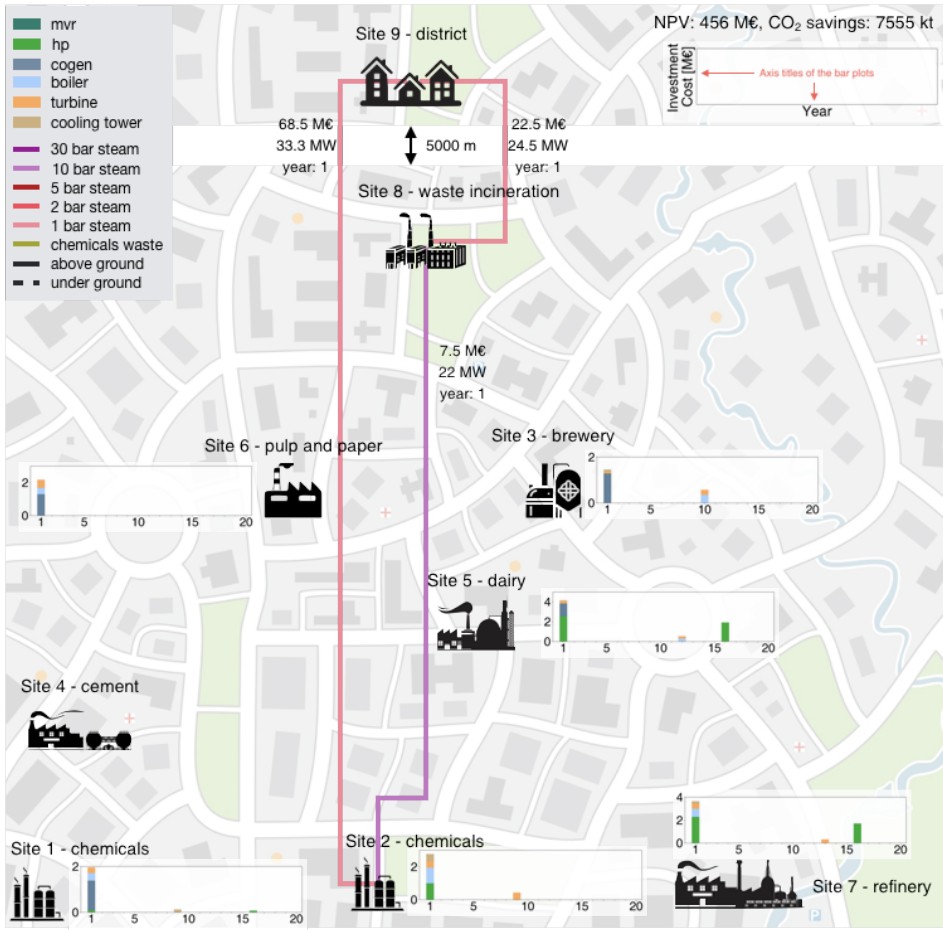

**Figure 7.** Optimal investment planning considering seasonality in district energy demand.

The impact of seasonality can also be observed in the investments in energy conversion technologies. Transferring most of the excess heat below the pinch point to the district, Site 2 has a lower potential for heat pump integration. Moreover, larger investment on boilers occur on Site 6 and 7, as they no longer receive excess heat from Site 8. In terms of investment timing, the results are

similar to those in Section 5.1; most of the investments occur in the first year, and the rest are for the repurchasing of equipment that has reached the end of its life span.

*5.3. Budget and Investment Constraints*

In real industrial retrofit projects, there is always a limitation on the budget, as the companies involved do not have unlimited resources. In such cases, it is important to spot the investments that are the most profitable under the project budget. The budget limitation is studied by introducing a constraint that limits the investments to 75% of the total investment cost of the optimal solution obtained in Section 5.1 (i.e., 80 M€). In addition, further constraints are applied to limit the yearly investment.

As a first case, an investment period of five years is considered. This means that all the investment decisions are taken in the first five years and the system is operated for the rest of the time given those decisions. It is assumed that the budget is evenly distributed within the investment period (i.e., 16 M€/year), under the condition that if it is not completely spent in a year, it can be transferred to the following one. With the investment constraints, NPV and $CO_2$ savings of the system decrease by 5% and 9%, respectively, compared to the optimal solution in Figure 6. Figure 8 illustrates the investment decisions and their corresponding year for the optimal solution with five-year investment horizon. Compared to Figure 6, the type of the technologies and equipment invested in are similar; cogeneration engines are installed in sites 1, 3, 5 and 6; heat pumps are installed in sites 2 and 7; boilers and turbines are replaced in almost all sites; and steam pipes are installed between sites 1, 5, 8 and 9. The impact of the budget restrictions can be seen in the timing of the investments as well as the size of some of the equipment; instead of purchasing most of the equipment in the first year, investments are spread over five years. In some of the years (e.g., year 1), the budget allowance is not fully used, either to be able to transfer some of it to the following year or because it is not sufficient enough for further investment. This way, large investments such as piping between Sites 8 and 9, which requires larger investments than the yearly allowance, are still possible. However, very large investments, e.g., 68.6 M€ piping between Sites 2 and 9 (see Figure 6) are not selected, as other options lead to more beneficial results for the objective function.

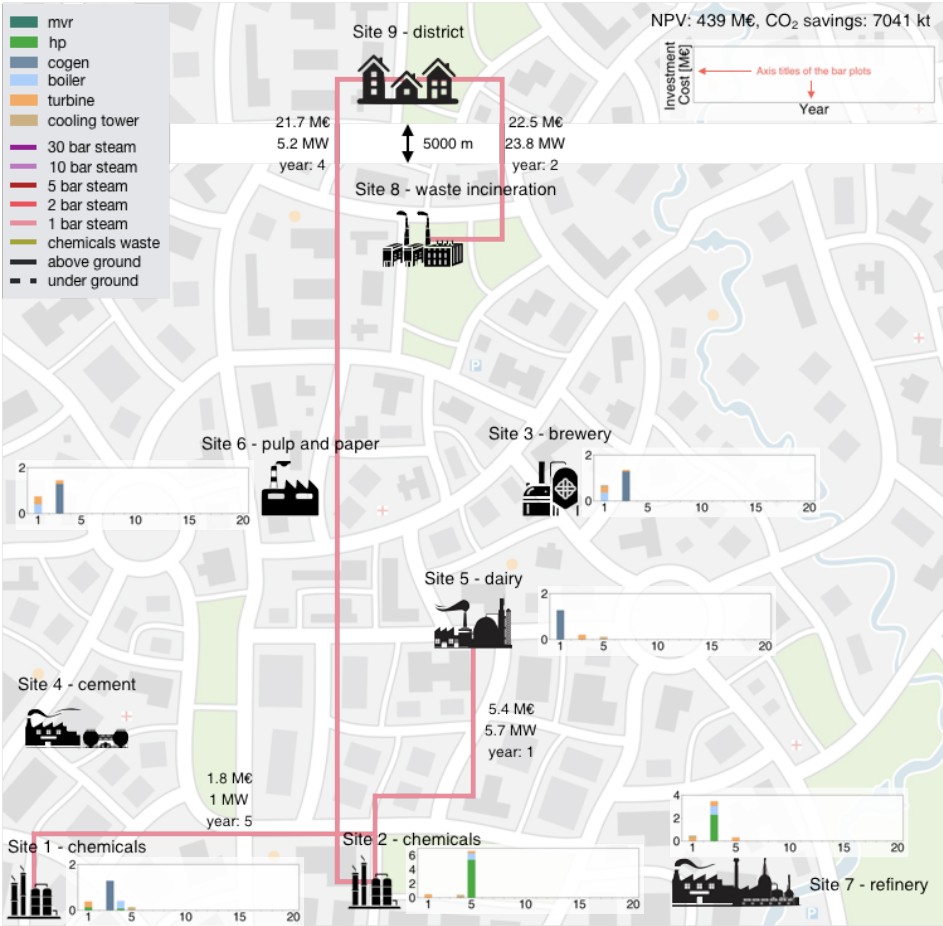

**Figure 8.** Optimal investment decisions with 5 years of investment and 16 M€ annual budget.

As a more conservative investment strategy, a case with a ten-year investment period is evaluated in Figure 9. As the investment period is broader, the yearly budget reduces to 8 M€, which results in a 13% lower NPV and 12% lower $CO_2$ reduction compared to the optimal solution in Figure 6. Similar to the previous case, it is assumed that the yearly investment budget, if unused, can be transferred to the following years. As the annual budget is reduced, the number of simultaneous investments decreases. The energy conversion system investments are prioritised over piping as they are smaller and can therefore be be completed earlier. Most of the intra-plant improvements via investing on better energy conversion systems are carried out in the first year. In the second year, the largest investment is in the pipeline between sites 2 and 5, as it is within the yearly budget. Following this, large investments are avoided for two years, to accumulate sufficient budget for piping between sites 8 and 9 (taking place in year five). Similarly, between year six and eight, investments are not made so as to accumulate sufficient budget for the large piping investment between sites 2 and 9 in year nine.

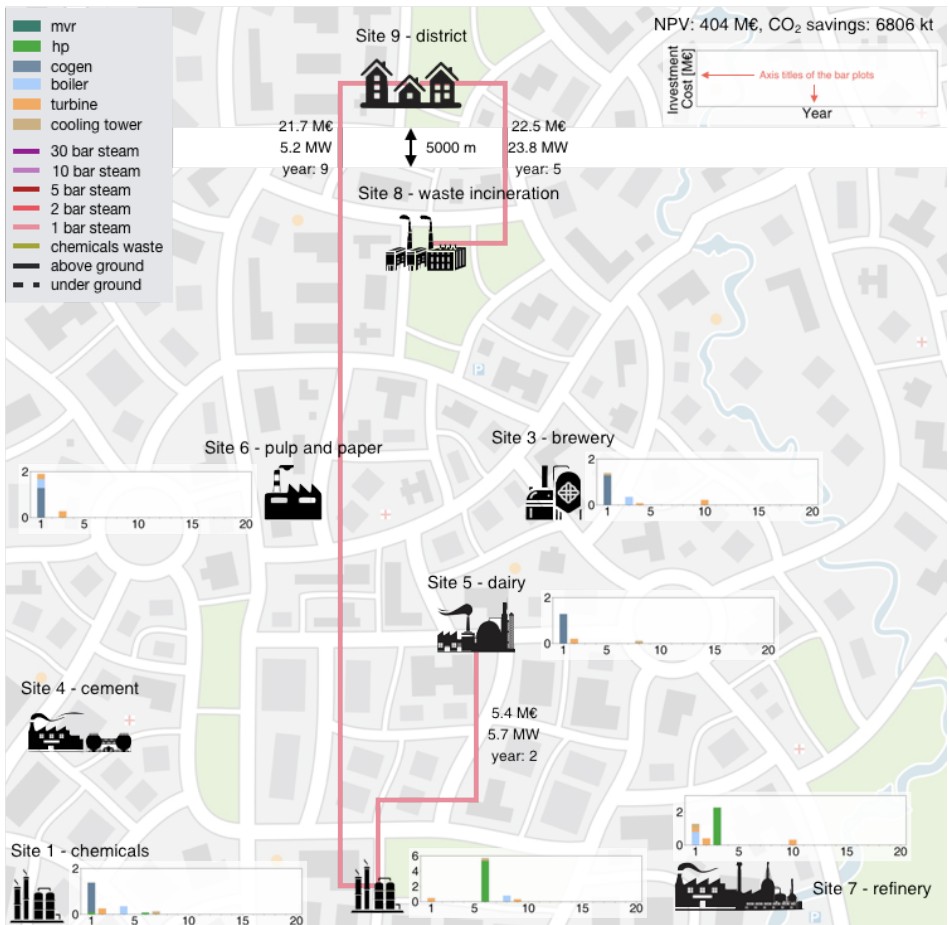

**Figure 9.** Optimal investment decisions with 10 years of investment and 8 M€ annual budget.

### 5.4. Multiple Scenarios for Investment

The parametric optimisation strategy described in Section 3.5 is utilised to obtain 30 solutions with different limits on the piping cost, representing multiple scenarios for investment. Figure 10 depicts the multicriteria comparison of the solutions from parametric optimisation as well as the one in Figure 9, all with 80 M€ overall investment budget and investment period of first ten years. The solutions are sorted with respect to NPV, which is the main objective and the solution in Figure 9, is highlighted with a bold line.

The solutions with high NPV also yield high $CO_2$ savings, taking the advantage of reduced operating cost (i.e., natural gas and electricity consumption). The solutions with low limit on piping investment budget rank the worst in NPV, $CO_2$ savings, operating cost and utility system investment. Conversely, piping investment does not always bring operational benefits which results in the solutions at the upper end of "Piping investment" axis having lower NPV than the ones below them. Heat shared with the district and industries has an inverse relationship, as the quantity of heat is limited and only its distribution varies between solutions. The solutions in which industrial excess heat is shared with the district yield better results in terms of NPV as natural gas and electricity prices are higher for the residential users compared to industries. The solution from Figure 9 ranks worse than half of the solutions obtained with parametric optimisation in both economic and environmental key performance indicators (KPIs). This can be explained by the use of a larger optimality gap (i.e., 5%) as the solution time is longer, and thus the solver does not try to explore better solutions. Despite having a higher optimality gap, the solution from Figure 9 requires computation time more than ten times that of the solutions from parametric optimisation.

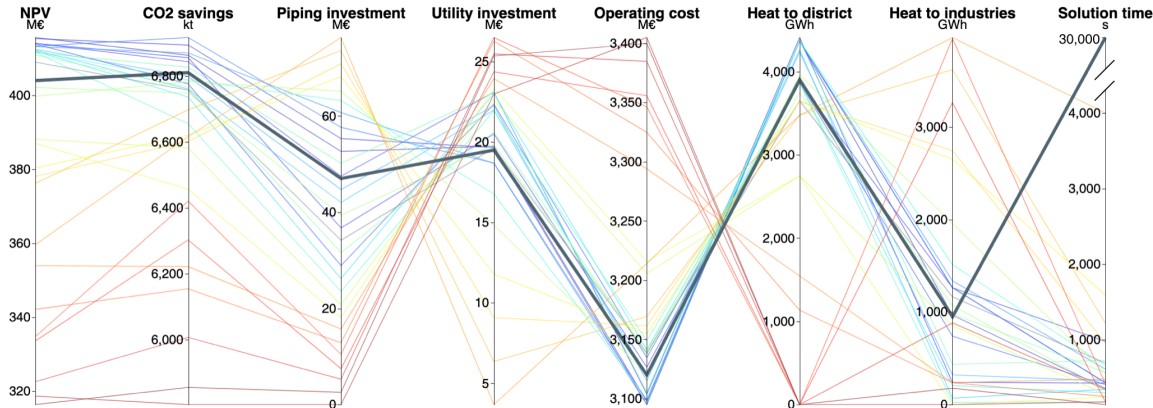

**Figure 10.** Multiple solutions generated by parametric optimisation; the higher the NPV, the colder the line colour.

*5.5. Comparison with Baseline*

Comparison of the baseline of the system with the optimal solutions identified (i.e., with and without budget constraints) is depicted in Figure 11. Baseline represents the current state of the system when the plants are operated with the energy conversion technologies that already exist on the sites. The investment cost in this case is required for the equipment reaching the end of their life span, to be able to continue the plant operation. Therefore, the operating cost remains constant throughout the twenty years, as nothing is done to improve the system efficiency. Similarly, in the optimal solution without budget constraints, the operating cost is the same for the span of the project, due to the fact that all the investments improving the system are carried out at the beginning of the first year. This also explains the large investment and 27% reduction in the operating cost in the first year compared to the baseline. When investments are limited to the first five years, the operating cost gradually improves 16–26% with the investments performed each year and then stabilises at the fifth year until the rest of the project. The same phenomenon happens for the case with ten years of investment; the operating cost improves by 16–24% in the investment period and then stays constant for the last ten years. Considering NPV and environmental impact, optimal investment planning without budget constraints improves the system by ~463 M€ and 7748 kt $CO_2$ in a twenty-year horizon, whereas investment budget constraints of five and ten years results in 5% and 13% lower NPV, and 9% and 12% lower $CO_2$ savings, respectively, when compared to the unconstrained solution. Although the investment planning strategy results in large investments, totalling 107 M€, yearly operating cost savings surpass 50 M€, resulting in a simple payback time of slightly greater than two years.

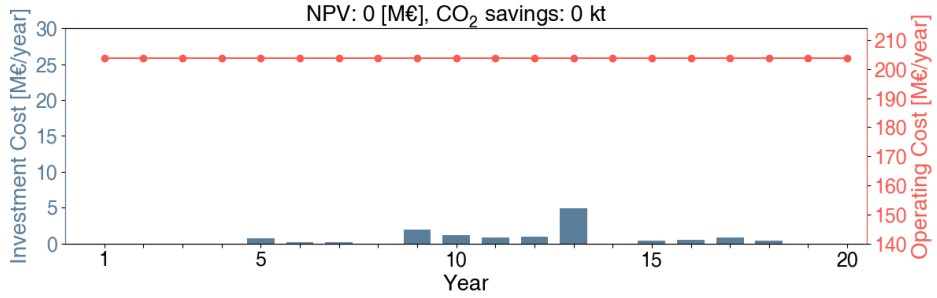

(**a**) Business as usual operation and investments.

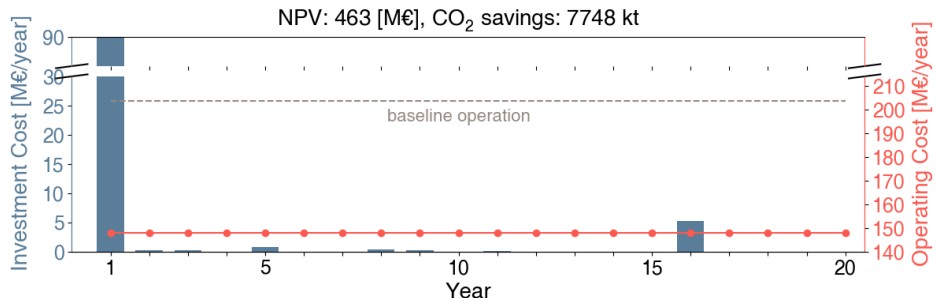

(**b**) Optimal solution without budget constraints.

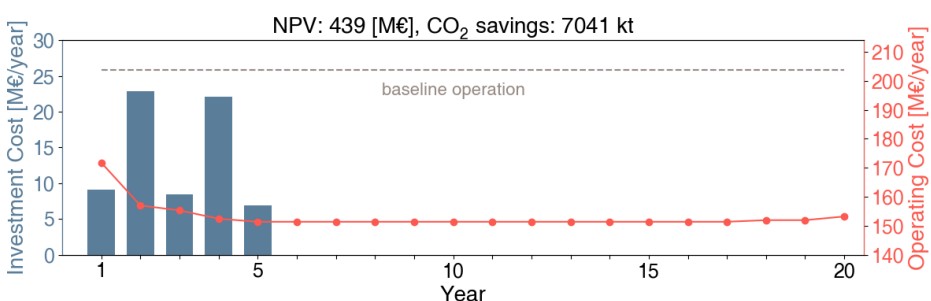

(**c**) Optimal solution with 5 years of investment and 16 M€/year budget.

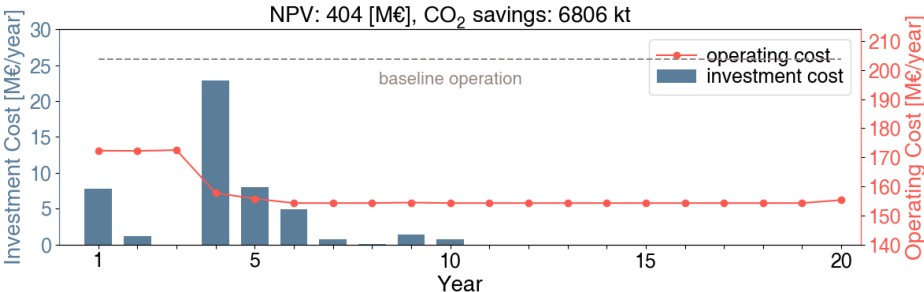

(**d**) Optimal solution with 10 years of investment and 8 M€/year budget.

**Figure 11.** Comparison of operating and investment costs of the optimal solutions with the baseline.

## 6. Conclusions

This work proposes an MILP framework, PIIP, which combines the efforts in process integration and long-term investment planning. The method takes advantage of PI, by modelling the energy and resource flows in detail, including heat cascade, mass and energy balances. A novel investment planning formulation is proposed and integrated with PI, capturing all investment actions, such

as commissioning and decommissioning of utility systems, while considering external exchanges via pipeline.

The method is applied to a large case study, with eight industrial plants from different sectors and a district neighbouring an industrial cluster, for a time horizon of twenty years. When the investment budget and period are not limited, all investments improving the operating cost are made in the first year, to maximise benefits from the operating cost savings as long as possible. Heat pumps and cogeneration engines are preferred over industrial boilers in all sites where their integration is possible. In inter-plant exchanges, priority is given to heat sharing with the district, as this option is more profitable because of lower industrial energy prices relative to residential ones. In addition, heat sharing over long distances with a single pipeline is preferred over investing in multiple pipelines connecting smaller nearby sites. When seasonality is taken into account, given the variations in the district demand, the investment decisions change drastically. A larger amount of heat is shared between the industrial cluster and the district resulting in less heat sharing between the industrial plants. Based on interactions between plants, different investment options become favourable. Thus, in the cases where energy demand varies greatly throughout a year, it is crucial to consider seasonality, to obtain the optimal selection and planning of the investments.

To simulate a more industrially-realistic scenario (i.e., refurbishment planning of plant infrastructure), an investment budget is imposed and the investment period is restricted to the first five or ten years, with the possibility of transferring budget from one year to the next. As the yearly budget does not allow for large early investments, they occur over the whole investment period. Some large investments found in the optimal solution without budget restrictions no longer appear, as annual budgets would need to accumulate for several years, making other solutions more attractive. As the investment strategy becomes more conservative (i.e., lower annual budget in a longer investment period), competition between investment in energy conversion technologies and inter-plant steam pipes increases, as parallel investments are not feasible. Although in the studied case, investment in energy conversion technologies receives priority over pipelines, the solution is dependent on the case study, energy profiles, prices and distances.

A strategy is proposed to generate multiple investment options using parametric optimisation by setting an upper limit for piping investment and varying it, and the results are compared with a single optimal solution. The parametric optimisation strategy not only generates 30 solutions in shorter time, but also finds solutions with better economic and environmental KPIs. Thus, in the case of industrial applications, it is better to generate multiple optimal solutions instead of trying to reach the global optimum. In all solutions ranking highly in economic and environmental KPIs, industrial excess heat is shared with the district. Thus, it is crucial to consider symbiosis options with a nearby heat consumers in industrial retrofit applications, which is often overlooked.

The optimal solution using the proposed method without investment restrictions leads to a ~463 M€ increase in the NPV of the system and 7748 kt $CO_2$ savings compared to the baseline, owing to operating cost benefits of investment decisions. Applying a budget limit on the investment cost with an investment period of five and ten years results in 5% and 13% decrease in NPV and 9% and 12% decrease in $CO_2$ savings compared to the solution without any budget limitation but they still provide significant improvement compared to the baseline. Although more conservative investment planning strategies result in slightly lower savings in the operating cost, they still lead to reductions of ~50 M€, or, in other words, a payback time of less than two years for the investments.

The method presented in this work provides a holistic strategy for investment planning of large industrial cases in long time horizons. Additional constraints can easily be integrated to customise it according to the limitations of the industrial clusters on the investment budget and periods. Future work includes adding stochasticity in the energy prices and cost of energy conversion technologies, as well as in the production capacity of plants and population of districts nearby. In addition, the objective function can be modified to optimise for an environmental objective instead of using a purely economic one.

**Author Contributions:** Conceptualisation, H.B., I.K. and F.M.; methodology, H.B.; software, H.B.; validation, H.B. and I.K.; formal analysis, H.B. and I.K.; investigation, H.B., I.K. and F.M.; resources, F.M.; data curation, H.B.; writing–original draft preparation, H.B.; writing–review and editing, H.B. and I.K.; visualisation, H.B.; supervision, I.K. and F.M.; project administration, I.K. and F.M.; funding acquisition, F.M.

**Funding:** This research project is financially supported by the Swiss Innovation Agency Innosuisse and is part of the Swiss Competence Center for Energy Research SCCER EIP. This project has received funding from the European Union's Horizon 2020 research and innovation programme under grant agreement No 679386. This work was supported by the Swiss State Secretariat for Education, Research and Innovation (SERI) under contract number 15.0217.

**Conflicts of Interest:** The authors declare no conflict of interest.

**Appendix A**

*Decommissioning Size Linearisation*

Decommissioning size is the product of binary and continuous variables as given in Equations (13) and (14) which are linearised with a set of constraints in Equations (A1)–(A3) for selling and Equations (A4)–(A6)

$$f_{u,p}^s \geq f_{u,p-1}^e - (1 - z_{u,p}^s) \cdot \mathrm{F}_u^{\max} \quad \forall\, u \in \mathbf{IU},\ p \in \mathbf{P} : p \neq 1 \tag{A1}$$

$$f_{u,p}^s \leq f_{u,p-1}^e \quad \forall\, u \in \mathbf{IU},\ p \in \mathbf{P} : p \neq 1 \tag{A2}$$

$$f_{u,p}^s \leq z_{u,p}^s \cdot \mathrm{F}_u^{\max} \quad \forall\, u \in \mathbf{IU},\ p \in \mathbf{P} : p \neq 1 \tag{A3}$$

$$f_{u,p}^d \geq f_{u,p-1}^e - (1 - z_{u,p}^d) \cdot \mathrm{F}_u^{\max} \quad \forall\, u \in \mathbf{IU},\ p \in \mathbf{P} : p \neq 1 \tag{A4}$$

$$f_{u,p}^d \leq f_{u,p-1}^e \quad \forall\, u \in \mathbf{IU},\ p \in \mathbf{P} : p \neq 1 \tag{A5}$$

$$f_{u,p}^d \leq z_{u,p}^d \cdot \mathrm{F}_u^{\max} \quad \forall\, u \in \mathbf{IU},\ p \in \mathbf{P} : p \neq 1 \tag{A6}$$

*Piping Investment Planning Model*

As pipes cannot be sold and have a longer life span than the other equipment, the investment decisions for them reduce to buying them or not in a given period $p \in \mathbf{P}$. Thus compared to the units, the investment planning model of pipes is simplified to the following set of rules:

- A pipe exists only if it has been purchased. See Equation (A7) for heat pipes and Equation (A11) for resource pipes;
- A pipe can be purchased only once. See Equation (A8) for heat pipes and Equation (A12) for resource pipes;
- A pipe can be used as long as its life span. See Equation (A9) for heat pipes and Equation (A13) for resource pipes;
- A pipe can be used only if it exists. See Equation (A10) for heat pipes and Equation (A14) for resource pipes;

$$z_{ly,sp,tr,o,p}^{eh} = \sum_{pp=1..p-1} z_{ly,sp,tr,o,pp}^{bh} \quad \forall\, ly \in \mathbf{L},\ sp \in \mathbf{SP},\ tr \in \mathbf{TR},\ o \in \mathbf{OL},\ p \in \mathbf{P} \tag{A7}$$

$$\sum_{p \in \mathbf{P}} z_{ly,sp,tr,o,p}^{bh} \leq 1 \quad \forall\, ly \in \mathbf{L},\ sp \in \mathbf{SP},\ tr \in \mathbf{TR},\ o \in \mathbf{OL} \tag{A8}$$

$$\sum_{p \in \mathbf{P}} y_{ly,sp,tr,o,p}^{ph} \leq 50 \quad \forall\, ly \in \mathbf{L},\ sp \in \mathbf{SP},\ tr \in \mathbf{TR},\ o \in \mathbf{OL} \tag{A9}$$

$$y_{ly,sp,tr,o,p}^{ph} \leq z_{ly,sp,tr,o,p}^{eh} \quad \forall\, ly \in \mathbf{L},\ sp \in \mathbf{SP},\ tr \in \mathbf{TR},\ o \in \mathbf{OL},\ p \in \mathbf{P} \tag{A10}$$

$$z_{ly,lc,o,u,p}^{er} = \sum_{pp=1..p-1} z_{ly,lc,o,u,pp}^{br} \quad \forall\, ly \in \mathbf{L},\ lc \in \mathbf{LC},\ o \in \mathbf{OL},\ u \in \mathbf{U}_{l,lc},\ p \in \mathbf{P} \tag{A11}$$

$$\sum_{p\in\mathbf{P}} z^{br}_{ly,lc,o,u,pp} \le 1 \quad \forall\, ly \in \mathbf{L},\ lc \in \mathbf{LC},\ o \in \mathbf{OL},\ u \in \mathbf{U}_{l,lc} \tag{A12}$$

$$\sum_{p\in\mathbf{P}} y^{pr}_{ly,lc,o,u,pp} \le 50 \quad \forall\, ly \in \mathbf{L},\ lc \in \mathbf{LC},\ o \in \mathbf{OL},\ u \in \mathbf{U}_{l,lc} \tag{A13}$$

$$y^{pr}_{ly,lc,o,u,pp} \le z^{er}_{ly,lc,o,u,p} \quad \forall\, ly \in \mathbf{L},\ lc \in \mathbf{LC},\ o \in \mathbf{OL},\ u \in \mathbf{U}_{l,lc},\ p \in \mathbf{P} \tag{A14}$$

*Piping cost calculations*

The cost of the pipes for heat ($C^{pipe_h}_{p,o,tr}$) and resource sharing ($C^{pipe_r}_{l,u,o}$) is calculated according to Equations (A15) and (A16).

$$C^{pipe_h}_{sp,o,tr} = \sum_{ps\in\mathbf{PS}} c^{pipe}_{ps} \cdot \mathrm{tf} \cdot l^{pipe}_{lc,o} \cdot n^{h}_{sp,o,tr,ps} \quad \forall\, sp \in \mathbf{SP},\ lc \in \mathbf{LC},\ o \in \mathbf{OL}_{lc},\ tr \in \mathbf{TR} \tag{A15}$$

$$C^{pipe_r}_{ly,u,o} = \sum_{ps\in\mathbf{PS}} c^{pipe}_{ps} \cdot \mathrm{tf} \cdot l^{pipe}_{lc,o} \cdot n^{r}_{ly,u,o,ps} \quad \forall\, ly \in \mathbf{L},\ u \in \mathbf{U}_l,\ lc \in \mathbf{LC},\ o \in \mathbf{OL}_{lc} \tag{A16}$$

where $n^{h}_{p,o,tr,ps}$ and $n^{r}_{l,u,o,ps}$ are binary variables deciding what size of pipe is used for heat and resource sharing respectively, tf is the trenching cost factor which is 1 for above-ground pipes (i.e. no trenching) and 1.3 for under-ground pipes [52] and $c^{pipe}_{ps}$ is the specific piping cost of the corresponding pipe size. Further details on piping cost calculations can be found in [21].

The specific piping cost is calculated based on the piping cost functions available in the literature [53–56]. Standard piping diameters and their corresponding cost are depicted in Table A1.

**Table A1.** Piping cost for standard piping diameters.

| Standard pipe size | 1 | 2 | 3 | 4 | 5 | 6 | 7 | 8 | 9 | 10 | 11 | 12 |
|---|---|---|---|---|---|---|---|---|---|---|---|---|
| Diameter (mm) | 20 | 40 | 80 | 100 | 200 | 300 | 400 | 500 | 600 | 800 | 1000 | 1500 |
| Specific cost (€/m) | 96 | 166 | 312 | 387 | 775 | 1180 | 1588 | 2008 | 2434 | 3304 | 4192 | 6474 |

*Budget constraints*

Equations (A17) and (A18) constraint the overall and annual investment costs according to available budget.

$$\sum_{p\in\mathbf{P}} C^{inv}_p \le c^{ob} \tag{A17}$$

$$C^{inv}_p \le c^{ab} \quad \forall\, p \in \mathbf{P} \tag{A18}$$

where $c^{ob}$ and $c^{ab}$ are overall and annual investment budgets respectively. When transferring the investment budget to the following year is allowed, Equations (A18) is replaced with Equations (A19)–(A21).

$$C^{tb}_p \le c^{ab} \quad \forall\, p \in \mathbf{P} : p = 1 \tag{A19}$$

$$C^{tb}_p = c^{ab} + C^{tb}_{p-1} - C^{inv}_{p-1} \quad \forall\, p \in \mathbf{P} : p \neq 1 \tag{A20}$$

$$C^{inv}_p = C^{tb}_p \quad \forall\, p \in \mathbf{P} \tag{A21}$$

where $C^{tb}_p$ is a continuous variable which decides how much of a yearly budget is transferred to the following year.

*Materials and engineering cost linearisation*

The products of binary and continuous variables in Equations (31)–(35) are linearised in Equations (A22)–(A24) for materials cost and Equations (A25)–(A27) for engineering cost.

$$C_{u,p}^{mt} \geq C_{u,p}^{b} \cdot F_u^{mt} - z_{u,p}^{bb} \cdot c_u^{max} \quad \forall\, u \in \mathbf{NU},\; p \in \mathbf{P} \tag{A22}$$

$$C_{u,p}^{mt} \leq C_{u,p}^{b} \cdot F_u^{mt} \quad \forall\, u \in \mathbf{NU},\; p \in \mathbf{P} \tag{A23}$$

$$C_{u,p}^{mt} \leq \left(1 - z_{u,p}^{bb}\right) \cdot c_u^{max} \quad \forall\, u \in \mathbf{NU},\; p \in \mathbf{P} \tag{A24}$$

$$C_{u,p}^{en} \geq C_{u,p}^{b} \cdot F_u^{en} - z_{u,p}^{bb} \cdot c_u^{max} \quad \forall\, u \in \mathbf{NU},\; p \in \mathbf{P} \tag{A25}$$

$$C_{u,p}^{en} \leq C_{u,p}^{b} \cdot F_u^{en} \quad \forall\, u \in \mathbf{NU},\; p \in \mathbf{P} \tag{A26}$$

$$C_{u,p}^{en} \leq \left(1 - z_{u,p}^{bb}\right) \cdot c_u^{max} \quad \forall\, u \in \mathbf{NU},\; p \in \mathbf{P} \tag{A27}$$

where $c_u^{max}$ is the maximum purchase cost of a unit which is used as a big M in the equations.

*Selling and dying value linearisation*

The product of binary and continuous variables in Equations (43)–(44) is linearised in Equations (A28)–(A30) for selling value and Equations (A31)–(A33) for dying value.

$$C_{u,p}^{s} \geq C_{u,p}^{rv} - \left(1 - z_{u,p}^{s}\right) \cdot c_u^{max} \quad \forall\, u \in \mathbf{IU},\; p \in \mathbf{P} \tag{A28}$$

$$C_{u,p}^{s} \leq C_{u,p}^{rv} \quad \forall\, u \in \mathbf{IU},\; p \in \mathbf{P} \tag{A29}$$

$$C_{u,p}^{s} \leq z_{u,p}^{s} \cdot c_u^{max} \quad \forall\, u \in \mathbf{IU},\; p \in \mathbf{P} \tag{A30}$$

$$C_{u,p}^{d} \geq C_{u,p}^{rv} - \left(1 - z_{u,p}^{d}\right) \cdot c_u^{max} \quad \forall\, u \in \mathbf{IU},\; p \in \mathbf{P} \tag{A31}$$

$$C_{u,p}^{d} \leq C_{u,p}^{rv} \quad \forall\, u \in \mathbf{IU},\; p \in \mathbf{P} \tag{A32}$$

$$C_{u,p}^{d} \leq z_{u,p}^{d} \cdot c_u^{max} \quad \forall\, u \in \mathbf{IU},\; p \in \mathbf{P} \tag{A33}$$

*Linearisation of the max function*

The max function in Equations (46) is linearised in Equations (A34)–(A38)

$$C_{u,p}^{s} \geq C_{u,p}^{sv} \quad \forall\, u \in \mathbf{IU},\; p \in \mathbf{P} \tag{A34}$$

$$C_{u,p}^{s} \geq z_{u,p-1}^{e} \cdot c_u^{sal} \quad \forall\, u \in \mathbf{IU},\; p \in \mathbf{P} : p \neq 1 \tag{A35}$$

$$C_{u,p}^{s} \leq C_{u,p}^{sv} + \left(1 - n_{u,p}^{rem}\right) \cdot c_u^{max} \quad \forall\, u \in \mathbf{IU},\; p \in \mathbf{P} \tag{A36}$$

$$C_{u,p}^{s} \leq z_{u,p-1}^{e} \cdot c_u^{sal} + \left(1 - n_{u,p}^{sal}\right) \cdot c_u^{max} \quad \forall\, u \in \mathbf{IU},\; p \in \mathbf{P} \tag{A37}$$

$$n_{u,p}^{rem} + n_{u,p}^{sal} = 1 \quad \forall\, u \in \mathbf{IU},\; p \in \mathbf{P} \tag{A38}$$

where $n_{u,p}^{rem}$ is a binary variable which takes the value of 1 if $C_{u,p}^{sv}$ is greater than $c_u^{sal}$ and $n_{u,p}^{sal}$ is binary variables which takes the value of 1 otherwise.

*Grand composite curves of the sites*

The thermal profiles of the industrial sites and the district are depicted in Figure A1 and Figure A2 respectively in the form of GCCs.

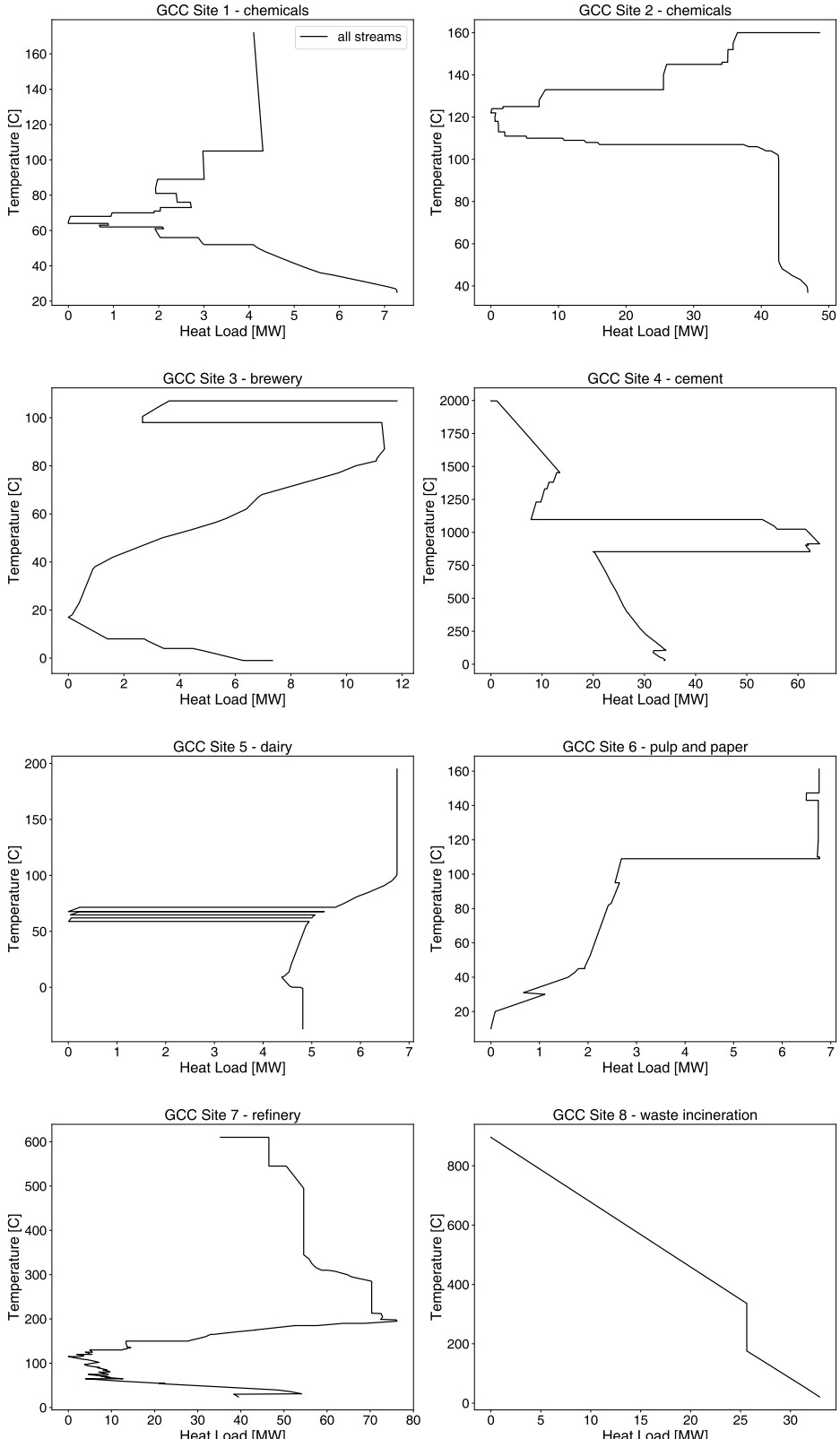

**Figure A1.** GCCs of the industrial processes in the case study.

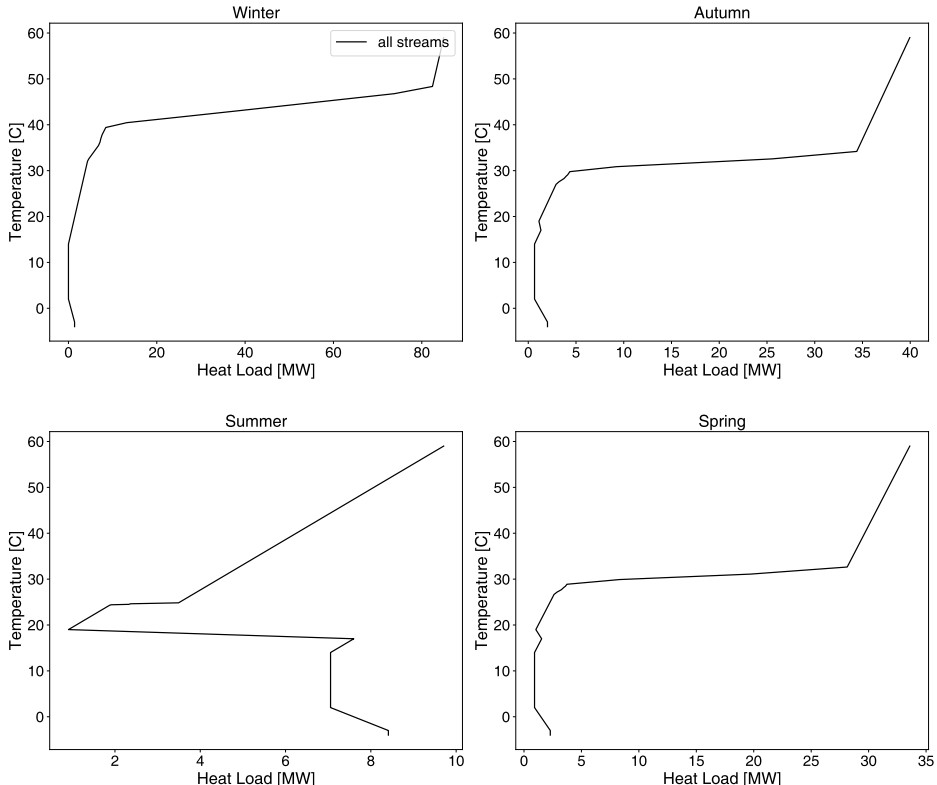

**Figure A2.** GCCs of the district in four seasons.

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
