# Peer review of "An Optimisation Approach for Long-Term Industrial Investment Planning"

_energies, doi:10.3390/en12214076_

Round 1
Reviewer 1 Report
The article proposes a method in order to incorporate the investment planning over long time horizons in the framework of process integration.
The main issue addressed by this research is innovative, relevant and interesting.
The article is overall accurately written and well detailed, although the analysis of the state of the art could be deepened.
Please find below my suggestions for improvements and amendments which can increase the completeness and readability of the paper.
IntroductionIn the introduction, you should mention also the steel industry as a big consumer of energy and producer of CO2 emission. This field is also relevant from the methodological point of view, as advanced single and multi-objective optimization techniques have been intensively applied. For instance, the following papers could be mentioned:
Nabernegg, S. et al., The deployment of low carbon technologies in energy intensive industries: A macroeconomic analysis for Europe, China and India (2017) Energies, 10 (3) Maddaloni A. et al: “Multi-objective optimization applied to retrofit analysis: A case study for the iron and steel industry,” (2015) Applied Thermal Engineering, 91, pp. 638-646. Lechtenböhmer, S. et al. Re-industrialisation and low-carbon economy-can they go together? Results from stakeholder-based scenarios for energy-intensive industries in the German state of North Rhine Westphalia, (2015) Energies, 8 (10), pp. 11404-11429.Moreover, please update the reference of the energy consumption with some more recent references (ref. [1]).
State of the artThe state of the art is too much focused on the investment planning. Since the work presented in this paper combines the strength of investment planning and PI, proper consideration should be given to both the aspects (investment planning and process integration). For instance, You could mention the following works:
Otto, A. et al. Power-to-steel: Reducing CO2 through the integration of renewable energy and hydrogen into the German steel industry, (2017) Energies, 10 (4). Porzio, G.F. et al., Process integration in energy and carbon intensive industries: an example of exploitation of optimization techniques and decision support, (2014) Applied Thermal Engineering, 70 (1), . Klemeš, J.J. et al. New directions in the implementation of Pinch Methodology (PM), (2018) Renewable and Sustainable Energy Reviews, 98, pp. 439-468. Kermani, M. et al., A holistic methodology for optimizing industrial resource efficiency, (2019) Energies, 12 (7) MethodIn order to increase the readability of the 3.1 paragraph, you should divide it into two subparagraphs: one related to the calculation of the first period and the other one related to the calculation of the other periods.
O the other hand, I suggest to merge the 3.4 and 3.5 paragraphs in one paragraph, since 3.4 is too short.
Case Studiescinv1 and cinv2: they refer to the investment cost, but, specifically, at what do they refer? Which are fixed/variable costs? Do the variable cost the same for all the sites?
Finit: how do you calculate this variable? At which equation does it refer?
Results and discussionIn this paragraph, you should introduce the several scenarios of the investment depicted in the subsequent paragraphs (5.1, 5.2,..).
Reviewer 2 Report
The article addresses an important issue in a setting that deserves much attention. The research framework is well organized, the research method is sound, and the findings are justifiable. My only concern is that the abstract might exceed the length limits.I would suggest the authors streamlining according to the journal policy. Currently the authors include too much information in the abstract.
Round 2
Reviewer 1 Report
The article proposes a method in order to incorporate the investment planning over long time horizons in the framework of process integration.
The main issue addressed by this research is innovative, relevant and interesting.
The article is overall accurately written and well detailed.
The authors improved the paper according to the suggestions of the reviewers. When such suggestions were not followed, a reasonable explanation has been provided. Therefore, I think that now the paper is suitable to the publication.